# Adaptive Data Analysis in a Balanced Adversarial Model

**Kobbi Nissim**
Department of Computer Science
Georgetown University
kobbi.nissim@georgetown.edu

**Uri Stemmer**
Blavatnik School of Computer Science
Tel Aviv University
Google Research
u@uri.co.il

**Eliad Tsfadia**
Department of Computer Science
Georgetown University
eliadtsfadia@gmail.com

## Abstract

In adaptive data analysis, a mechanism gets $n$ i.i.d. samples from an unknown distribution $\mathcal{D}$, and is required to provide accurate estimations to a sequence of adaptively chosen statistical queries with respect to $\mathcal{D}$. Hardt and Ullman [2014] and Steinke and Ullman [2015a] showed that, in general, it is computationally hard to answer more than $\Theta(n^2)$ adaptive queries, assuming the existence of one-way functions.

However, these negative results strongly rely on an adversarial model that significantly advantages the adversarial analyst over the mechanism, as the analyst, who chooses the adaptive queries, also chooses the underlying distribution $\mathcal{D}$. This imbalance raises questions with respect to the applicability of the obtained hardness results – an analyst who has complete knowledge of the underlying distribution $\mathcal{D}$ would have little need, if at all, to issue statistical queries to a mechanism which only holds a finite number of samples from $\mathcal{D}$.

We consider more restricted adversaries, called *balanced*, where each such adversary consists of two separate algorithms: The *sampler* who is the entity that chooses the distribution and provides the samples to the mechanism, and the *analyst* who chooses the adaptive queries, but has no prior knowledge of the underlying distribution (and hence has no a priori advantage with respect to the mechanism).

We improve the quality of previous lower bounds by revisiting them using an efficient *balanced* adversary, under standard public-key cryptography assumptions. We show that these stronger hardness assumptions are unavoidable in the sense that any computationally bounded *balanced* adversary that has the structure of all known attacks, implies the existence of public-key cryptography.

## 1 Introduction

Statistical validity is a widely recognized crucial feature of modern science. Lack of validity – popularly known as the *replication crisis* in science poses a serious threat to the scientific process and also to the public's trust in scientific findings.

One of the factors leading to the replication crisis is the inherent adaptivity in the data analysis process. To illustrate adaptivity and its effect, consider a data analyst who is testing a specific research

37th Conference on Neural Information Processing Systems (NeurIPS 2023).

hypothesis. The analyst gathers data, evaluates the hypothesis empirically, and often finds that their hypothesis is not supported by the data, leading to the formulation and testing of more hypotheses. If these hypotheses are tested and formed based on the same data (as acquiring fresh data is often expensive or even impossible), then the process is of *adaptive data analysis* (ADA) because the choice of hypotheses depends on the data. However, ADA no longer aligns with classical statistical theory, which assumes that hypotheses are selected independently of the data (and preferably before gathering data). ADA may lead to overfitting and hence false discoveries.

Statistical validity under ADA is a fundamental problem in statistics, that has received only partial answers. A recent line of work, initiated by Dwork et al. [2015c] and includes [Hardt and Ullman, 2014, Dwork et al., 2015a,b, Steinke and Ullman, 2015a,b, Bassily et al., 2016, Rogers et al., 2016, Russo and Zou, 2016, Smith, 2017, Feldman and Steinke, 2017, Nissim et al., 2018, Feldman and Steinke, 2018, Shenfeld and Ligett, 2019, Jung et al., 2020, Fish et al., 2020, Dagan and Kur, 2022, Kontorovich et al., 2022, Dinur et al., 2023, Blanc, 2023] has resulted in new insights into ADA and robust paradigms for guaranteeing statistical validity in ADA. A major objective of this line of work is to design optimal mechanisms M that initially obtain a dataset $\mathcal{S}$ containing $n$ i.i.d. samples from an unknown distribution $\mathcal{D}$, and then answers adaptively chosen queries with respect to $\mathcal{D}$. Importantly, all of M's answers must be accurate with respect to the underlying distribution $\mathcal{D}$, not just w.r.t. the empirical dataset $\mathcal{S}$. The main question is how to design an efficient mechanism that provides accurate estimations to adaptively chosen statistical queries, where the goal is to maximize the number of queries M can answer. This objective is achieved by providing both upper- and lower-bound constructions, where the lower-bound constructions demonstrate how an adversarial analyst making a small number of queries to an arbitrary M can invariably force M to err. The setting for these lower-bound proofs is formalized as a two-player game between a mechanism M and an adversary A as in Game 1.1.

---

**Game 1.1** (ADA game between a mechanism M and an adversarial analyst A).

- M *gets a dataset $\mathcal{S}$ of $n$ i.i.d. samples from an **unknown** distribution $\mathcal{D}$ over $\mathcal{X}$.*
- *For $i = 1, \ldots, \ell$:*
    - A *sends a query $q_i \colon \mathcal{X} \mapsto [-1, 1]$ to M.*
    - M *sends an answer $y_i \in [-1, 1]$ to A.*
      *(As A and M are stateful, $q_i$ and $y_i$ may depend on the previous messages.)*
    - M *fails if $\exists i \in [\ell]$ s.t. $|y_i - \mathrm{E}_{x \sim \mathcal{D}}[q_i(x)]| > 1/10$.*

---

A question that immediately arises from the description of Game 1.1 is to whom should the distribution $\mathcal{D}$ be unknown, and how to formalize this lack of knowledge. Ideally, the mechanism M should succeed with high probability for every unknown distribution $\mathcal{D}$ and against any adversary A.

In prior work, this property was captured by letting the adversary choose the distribution $\mathcal{D}$ at the outset of Game 1.1. Namely, the adversary A can be seen as a pair of algorithms $(A_1, A_2)$, where $A_1$ chooses the distribution $\mathcal{D}$ and sends a state st to $A_2$ (which may contain the entire view of $A_1$), and after that, M and $A_2(\mathsf{st})$ interacts in Game 1.1. In this adversarial model, Hardt and Ullman [2014] and Steinke and Ullman [2015a] showed that, assuming the existence of one way functions, it is computationally hard to answer more than $\Theta(n^2)$ adaptive queries. These results match the state-of-the-art constructions [Dwork et al., 2015c,a,b, Steinke and Ullman, 2015b, Bassily et al., 2016, Feldman and Steinke, 2017, 2018, Dagan and Kur, 2022, Blanc, 2023].[1] In fact, each such negative result was obtained by constructing a *single* adversary A that fails *all* efficient mechanisms. This means that, in general, it is computationally hard to answer more than $\Theta(n^2)$ adaptive queries even when the analyst's algorithm is known to the mechanism. On the other hand, in each of these negative results, the adversarial analyst has a significant advantage over the mechanism – their ability to select the distribution $\mathcal{D}$. This allows the analyst to inject random trapdoors in $\mathcal{D}$ (e.g., keys of an encryption scheme) which are then used in forcing a computationally limited mechanism to fail, as the mechanism does not get a hold of the trapdoor information.

---

[1]Here is an example of a mechanism that handles $\tilde{\Theta}(n^2)$ adaptive queries using differential privacy: Given a query $q_i$, the mechanism returns an answer $y_i = \frac{1}{n} \sum_{x \in \mathcal{S}} x + \nu_i$ where the $\nu_i$'s are independent Gaussian noises, each with standard deviation of $\tilde{O}(\sqrt{\ell}/n)$. The noises guarantee that the entire process is "private enough" for avoiding overfitting in the ADA game, and accuracy is obtained whenever $\ell = \tilde{O}(n^2)$.

For most applications, the above adversarial model seems to be too strong. For instance, a data analyst who is testing research hypotheses usually has no knowledge advantage about the distribution that the mechanism does not have. In this typical setting, even if the underlying distribution $\mathcal{D}$ happens to have a trapdoor, if the analyst recovers the trapdoor then the mechanism should also be able to recover it and hence disable its adversarial usage.

In light of this observation, we could hope that in a balanced setting, where the underlying distribution is unknown to both the mechanism and the analyst, it would always be possible for M to answer more than $O(n^2)$ adaptive queries. To explore this possibility, we introduce what we call a *balanced* adversarial model.

**Definition 1.2** (Balanced Adversary). *A* balanced *adversary* A *consists of two isolated algorithms: The* sampler $A_1$*, which chooses a distribution $\mathcal{D}$ and provides i.i.d. samples to the mechanism* M*, and the* analyst $A_2$*, which asks the adaptive queries. No information is transferred from $A_1$ to $A_2$. See Game 1.3.*

---

**Game 1.3** (The ADA game between a mechanism M and a balanced adversary $A = (A_1, A_2)$).

- $A_1$ *chooses a distribution $\mathcal{D}$ over $\mathcal{X}$ (specified by a sampling algorithm) and provides $n$ i.i.d. samples $\mathcal{S}$ to M (by applying the sampling algorithm $n$ times).*
  /* $A_1$ *does not provide $A_2$ with any information */*
- M *and $A_2$ play Game 1.1 (with respect to $\mathcal{D}$ and $\mathcal{S}$).*
  M *fails if and only if it fails in Game 1.1.*

---

Note that the difference between the balanced model and the previous (imbalanced) one is whether $A_1$ can send a state to $A_2$ after choosing the distribution $\mathcal{D}$ (in the imbalanced model it is allowed, in contrast to the balanced model).[2]

We remark that the main advantage of the balanced model comes when considering a publicly known sampler $A_1$ (as we do throughout this work). This way, $A_1$ captures the common knowledge that both the mechanism M and the analyst $A_2$ have about the underlying distribution $\mathcal{D}$.

**Question 1.4.** *Do the lower-bounds proved in prior work hold also for balanced adversaries?*

In this work we answer Question 1.4 in the positive. We do that using a publicly known analyst $A_2$ (which even makes it stronger than what is required for a lower bound). I.e., even though the sampler $A_1$ and the analyst $A_2$ are publicly known and cannot communicate with each other, they fail any computationally bounded mechanism. However, our lower-bound is based on stronger hardness assumptions than in prior work, namely, we use public-key cryptography.

## 1.1 Our Results

Our first result is a construction of a *balanced* adversary forcing any computationally bounded mechanim to fail in Game 1.3.

**Theorem 1.5** (Computational lower bound, informal). *There exists an efficient* balanced *adversary* $A = (A_1, A_2)$ *that fails any computationally bounded mechanism* M *using $\Theta(n^2)$ adaptive queries. Moreover, it does so by choosing a distribution over a small domain.*

Our construction in Theorem 1.5 uses the structure of previous attacks of Hardt and Ullman [2014] and Steinke and Ullman [2015a], but relies on a stronger hardness assumption of public-key cryptography. We prove that this is unavoidable.

**Theorem 1.6** (The necessity of public-key cryptography for proving Theorem 1.5, informal). *Any computationally bounded* balanced *adversary that follows the structure of all currently known attacks, implies the existence of public-key cryptography (in particular, a* key-agreement *protocol).*

In Section 1.3 we provide proof sketches of Theorems 1.5 and 1.6, where the formal statements appear in Sections 3 and 4 (respectively) and the formal proofs appear in the supplementary material.

---

[2]An additional (minor) difference is that we chose in our model to let $A_1$ also provide the i.i.d. samples to M. This is only useful for Theorem 1.6 as we need there that choosing $\mathcal{D}$ and sampling from $\mathcal{D}$ are both computationally efficient (which are simply captured by saying that $A_1$ is computationally bounded).

**Potential Consequences for the Information Theoretic Setting.** Theorem 1.6 has immediate implication to the information theoretic setting, and allow for some optimism regarding the possibility of constructing an inefficient mechanism that answers many adaptive queries.

It is known that an inefficient mechanism can answer exponentially many adaptive queries, but such results have a strong dependency on the domain size. For instance, the Private Multiplicative Weights algorithm of Hardt and Rothblum [2010] can answer $2^{\tilde{O}\left(n/\sqrt{\log|\mathcal{X}|}\right)}$ adaptive queries accurately. However, this result is not useful whenever $n \leq O(\sqrt{\log|\mathcal{X}|})$. Indeed, Steinke and Ullman [2015a] showed that this dependency is unavoidable in general, by showing that large domain can be used for constructing a similar, unconditional, adversary that fails any computationally unbounded mechanism after $\Theta(n^2)$ queries. Our Theorem 1.6 implies that such an attack cannot be implemented in the balanced setting, which gives the first evidence that there might be a separation between the computational and information theoretic setting under the balanced adversarial model (in contrast with the imbalanced model).

**Corollary 1.7.** *There is no* balanced *adversary that follows the structure of all currently known attacks, and fails any (computationally* unbounded*) mechanism.*

In order to see why Corollary 1.7 holds, suppose that we could implement such kind of attack using a *balanced* adversary. Then by Theorem 1.6, this would imply that we could construct an information-theoretic key agreement protocol (i.e., a protocol between two parties that agree on a key that is secret from the eyes of a computationally *unbounded* adversary that only sees the transcript of the execution). But since the latter does not exist, we conclude that such a *balanced* adversary does not exists either. In other words, we do not have a negative result that rules out the possibility of constructing an inefficient mechanism that can answer many adaptive queries of a *balanced* adversary, and we know that if a negative result exists, then by Theorem 1.6 it cannot follow the structure of Hardt and Ullman [2014], Steinke and Ullman [2015a].

## 1.2 Comparison with Elder [2016]

The criticism about the lower bounds of Hardt and Ullman [2014], Steinke and Ullman [2015a] is not new and prior work has attempted at addressing them with only partial success.

For example, Elder [2016] presented a similar "balanced" model (called "Bayesian ADA"), where both the analyst and the mechanism receive a *prior* $\mathcal{P}$ which is a family of distributions, and then the distribution $\mathcal{D}$ is drawn according to $\mathcal{P}$ (unknown to both the mechanism and the analyst).

From an information theoretic point of view, this model is equivalent to ours when the sampler $A_1$ is publicly known, since $A_1$ simply defines a prior. But from a computational point of view, defining the sampling process (i.e., sampling $\mathcal{D}$ and the i.i.d. samples from it) in an algorithmic way is better when we would like to focus on computationally bounded samplers.

Elder [2016] only focused on the information-theoretic setting. His main result is that a certain family of mechanisms (ones that only use the posterior means) cannot answer more than $\tilde{O}(n^4)$ adaptive queries. This, however, does not hold for any mechanism's strategy. In particular, it does not apply to general computationally efficient mechanisms. Our negative result is quantitatively stronger ($n^2$ vs $n^4$) and it applies for all computationally efficient mechanisms.[3]

Table 1 summarizes the comparison between Theorem 1.5 and the prior lower bounds (ignoring computational hardness assumptions).

## 1.3 Techniques

We follow a similar technique to that used in Hardt and Ullman [2014] and Steinke and Ullman [2015a], i.e., a reduction to a restricted set of mechanisms, called *natural*.

**Definition 1.8** (Natural mechanism [Hardt and Ullman, 2014]). *A mechanism* M *is* natural *if, when given a sample* $\mathcal{S} = (x_1, \ldots, x_n) \in \mathcal{X}^n$ *and a query* $q \colon \mathcal{X} \to [-1, 1]$, M *returns an answer that is a function solely of* $(q(x_1), \ldots, q(x_n))$. *In particular,* M *does not evaluate* $q$ *on other data points of its choice.*

---

[3] Our result is not directly comparable to that of Elder [2016], because our negative result does not say anything for non-efficient mechanisms, while his result does rule out a certain family of non-efficient mechanisms.

| | **Balanced?** | **Class of Mechanisms** | **# of Queries** | **Dimension** $(\log|\mathcal{X}|)$ |
|---|---|---|---|---|
| Steinke and Ullman [2015a] | No | PPT Algorithms | $\tilde{O}(n^2)$ | $n^{o(1)}$ |
| Steinke and Ullman [2015a] | No | All | $\tilde{O}(n^2)$ | $O(n^2)$ |
| Elder [2016] | Yes | Certain Family | $\tilde{O}(n^4)$ | $\tilde{O}(n^4)$ |
| Theorem 1.5 | Yes | PPT Algorithms | $\tilde{O}(n^2)$ | $n^{o(1)}$ |

**Table 1:** Comparison between the lower bounds for adaptive data analysis.

Hardt and Ullman [2014] and Steinke and Ullman [2015a] showed that there exists an adversarial analyst $\widetilde{\mathsf{A}}$ that fails any *natural* mechanism M, even when M is computationally unbounded, and even when $\mathcal{D}$ is chosen to be the uniform distribution over $\{1, 2, \ldots, m = 2000n\}$ (I.e., $\mathcal{D}$ is known to everyone). While general mechanisms could simply use the knowledge of the distribution to answer any query, *natural* mechanisms are more restricted, and can only provide answers based on the $n$-size dataset $\mathcal{S}$ that they get. The restriction to natural mechanisms allowed Steinke and Ullman [2015a] to use *interactive fingerprinting codes*, which enable to reveal $\mathcal{S}$ using $\Theta(n^2)$ adaptive queries when the answers are accurate and correlated with $\mathcal{S}$.

To construct an attacker A that fails any computationally bounded mechanism (and not just natural mechanisms), prior work forced the mechanism to behave naturally by using a private-key encryption scheme. More specifically, the adversary first samples $m$ secret keys $\mathsf{sk}_1, \ldots, \mathsf{sk}_m$, and then defines $\mathcal{D}$ to be the uniform distribution over the pairs $\{(j, \mathsf{sk}_j)\}_{j=1}^m$. The adversary then simulates an adversary $\widetilde{\mathsf{A}}$ which fails natural mechanisms as follows: a query $\tilde{q} \colon [m] \to [-1, 1]$ issued by $\widetilde{\mathsf{A}}$ is translated by A to a set of $m$ encryptions $\{\mathsf{ct}_j\}_{j=1}^m$ where each $\mathsf{ct}_j$ is an encryption of $\tilde{q}(j)$ under the key $\mathsf{sk}_j$. These encryptions define a new query $q$ that on input $(j, \mathsf{sk})$, outputs the decryption of $\mathsf{ct}_j$ under the key $\mathsf{sk}$. However, since M is computationally bounded and has only the secret keys that are part of its dataset $\mathcal{S}$, it can only decrypt the values of $\tilde{q}$ on points in $\mathcal{S}$, yielding that it effectively behaves *naturally*.

Note that the above attack A is *imbalanced* as it injects the secret keys $\mathsf{sk}_1, \ldots, \mathsf{sk}_m$ into $\mathcal{D}$ and then uses these keys when it forms queries. In other words, even though the attacker A is known to the mechanism, A is able to fail M by creating a secret correlation between its random coins and the distribution $\mathcal{D}$.

### 1.3.1 Balanced Adversary via Identity Based Encryption Scheme

For proving Theorem 1.5, we replace the private-key encryption scheme with a public-key primitive called *identity-based encryption* (IBE) scheme [Shamir, 1984, Cocks, 2001, Boneh and Franklin, 2001]. Such a scheme enables to produce $m$ secret keys $\mathsf{sk}_1, \ldots, \mathsf{sk}_m$ along with a master public key mpk. Encrypting a message to a speific identity $j \in [m]$ only requires mpk, but decrypting a message for identity $j$ must be done using its secret key $\mathsf{sk}_j$. Using an IBE scheme we can achieve a reduction to *natural* mechanisms via a *balanced* adversary $\mathsf{A} = (\mathsf{A}_1, \mathsf{A}_2)$ as follows: $\mathsf{A}_1$ samples keys $\mathsf{mpk}, \mathsf{sk}_1, \ldots, \mathsf{sk}_m$ according to the IBE scheme, and defines $\mathcal{D}$ to be the uniform distribution over the triplets $\{(j, \mathsf{mpk}, \mathsf{sk}_j)\}_{j=1}^m$. The analyst $\mathsf{A}_2$, which does not know the keys, first asks queries of the form $q(j, \mathsf{mpk}, \mathsf{sk}) = \mathsf{mpk}_k$ for every bit $k$ of mpk in order to reveal it. Then, it follows a strategy as in the previous section, i.e., it simulates an adversary $\widetilde{\mathsf{A}}$ which foils natural mechanisms by translating each query query $\tilde{q} \colon [m] \to [-1, 1]$ issued by $\widetilde{\mathsf{A}}$ by encrypting each $\tilde{q}(j)$ for identity $j$ using mpk. Namely, the IBE scheme allowed the analyst to implement the attack of Hardt and Ullman [2014] and Steinke and Ullman [2015a], but without having to know the secret keys $\mathsf{sk}_1, \ldots, \mathsf{sk}_m$.

We can implement the IBE scheme using a standard public-key encryption scheme: in the sampling process, we sample $m$ independent pairs of public and secret keys $\{(\mathsf{pk}_j, \mathsf{sk}_j)\}_{j=1}^m$ of the original scheme, and define $\mathsf{mpk} = (\mathsf{pk}_1, \ldots, \mathsf{pk}_m)$. When encrypting a message for identity $j \in [m]$, we could simply encrypt it using $\mathsf{pk}_j$ (part of mpk), which can only be decrypted using $\mathsf{sk}_j$. The disadvantage of this approach is the large master public key mpk that it induces. Applying the encryption scheme with security parameter of $\lambda$, the master key mpk will be of size $\lambda \cdot m$ and not just $\lambda$ as the sizes of the secret keys. This means that implementing our *balanced* adversary with such an encryption scheme would result with a distribution over a large domain $\mathcal{X}$, which would not rule out the possibility to construct a mechanism for distributions over smaller domains. Yet, Döttling and

Garg [2021] showed that it is possible to construct a fully secure IBE scheme using a small mpk of size only $O(\lambda \cdot \log m)$ under standard hardness assumptions (e.g., the *Computational Diffie Helman* problem [Diffie and Hellman, 1976][4] or the hardness of *factoring*).

### 1.3.2 Key-Agreement Protocol via Balanced Adversary

In order to prove Theorem 1.6, we first explain what type of adversaries the theorem applies to. Recall that in all known attacks (including ours), the adversary A wraps a simpler adversary $\widetilde{\mathsf{A}}$ that fails *natural* mechanisms. In particular, the wrapper A has two key properties:

1. A knows $\mathrm{E}_{x\sim\mathcal{D}}[q_\ell(x)]$ for the last query $q_\ell$ that it asks (becuase it equals to $\frac{1}{m}\sum_{j=1}^m \tilde{q}_\ell(j)$, where $\tilde{q}_\ell$ is the wrapped query which is part of A's view), and

2. If the mechanism attempts to behave accurately in the first $\ell - 1$ rounds (e.g., it answers the empirical mean $\frac{1}{n}\sum_{x\in\mathcal{S}} q(x)$ for every query $q$), then A, as a wrapper of $\widetilde{\mathsf{A}}$, will be able to ask a last query $q_\ell$ that would fail any computationally bounded last-round strategy for the mechanism.

We next show that any computationally bounded *balanced* adversary A that has the above two properties, can be used for constructing a key-agreement protocol. That is, a protocol between two computationally bounded parties $\mathsf{P}_1$ and $\mathsf{P}_2$ that enable them to agree on a value which cannot be revealed by a computationally bounded adversary who only sees the transcript of the execution. See Protocol 1.9.

---

**Protocol 1.9** (Key-Agreement Protocol $(\mathsf{P}_1, \mathsf{P}_2)$ via a *balanced* adversary $\mathsf{A} = (\mathsf{A}_1, \mathsf{A}_2)$)**.**

*Input:* $1^n$. Let $\ell = \ell(n)$ and $\mathcal{X} = \mathcal{X}(n)$ be the number queries and the domain that is used by the adversary A.

*Operation:*

- $\mathsf{P}_1$ emulates $\mathsf{A}_1$ on input $n$ for obtaining a distribution $\mathcal{D}$ over $\mathcal{X}$ (specified by a sampling procedure), and samples $n$ i.i.d. samples $\mathcal{S}$.
- $\mathsf{P}_2$ initializes an emulation of $\mathsf{A}_2$ on input $n$.
- For $i = 1$ to $\ell$:
    1. $\mathsf{P}_2$ receives the $i^{\mathrm{th}}$ query $q_i$ from the emulated $\mathsf{A}_2$ and sends it to $\mathsf{P}_1$.
    2. $\mathsf{P}_1$ computes $y_i = \frac{1}{n}\sum_{x\in\mathcal{S}} q_i(x)$, and sends it to $\mathsf{P}_2$.
    3. $\mathsf{P}_2$ sends $y_i$ as the $i^{\mathrm{th}}$ answer to the emulated $\mathsf{A}_2$.
- $\mathsf{P}_1$ and $\mathsf{P}_2$ agree on $\mathrm{E}_{x\sim\mathcal{D}}[q_\ell(x)]$.

---

The agreement of Protocol 1.9 relies on the ability of $\mathsf{P}_1$ and $\mathsf{P}_2$ to compute $\mathrm{E}_{x\sim\mathcal{D}}[q_\ell(x)]$. Indeed, $\mathsf{P}_1$ can accurately estimate it using the access to the sampling procedure, and $\mathsf{P}_2$ can compute it based on the view of the analyst $\mathsf{A}_2$ (follows by Property 1).

To prove the secrecy guarantee of Protocol 1.9, assume towards a contradiction that there exists a computationally bounded adversary G that given the transcript of the execution, can reveal $\mathrm{E}_{x\sim\mathcal{D}}[q_\ell(x)]$. Now consider the following mechanism for the ADA game: In the first $\ell - 1$ queries, answer the empirical mean, but in the last query, apply G on the transcript and answer its output. By the assumption on G, the mechanism will be able to accurately answer the last query, in contradiction to Property 2.

We note that Property 1 can be relaxed by only requiring that A is able to provide a "good enough" estimation of $\mathrm{E}_{x\sim\mathcal{D}}[q_\ell(x)]$. Namely, as long as the estimation provided in Property 1 is better than the estimation that an adversary can obtain in Property 2 (we prove that an $n^{\Omega(1)}$ multiplicative gap suffices), this would imply that Protocol 1.9 is a *weak* key-agreement protocol, which can be amplified to a fully secure one using standard techniques.

We also note that by requiring in Game 1.3 that $\mathsf{A}_1$ samples from $\mathcal{D}$ according to the sampling procedure, we implicitly assume here that sampling from $\mathcal{D}$ can be done efficiently (because $\mathsf{A}_1$ is

---

[4]CDH is hard with respect to a group $\mathbb{G}$ of order $p$, if given a random generator $g$ along with $g^a$ and $g^b$, for uniformly random $a, b \in [p]$, as inputs, the probability that a PPT algorithm can compute $g^{ab}$ is negligible.

assumed to be computationally bounded). Our reduction to key-agreement relies on this property, since if sampling from $\mathcal{D}$ could not be done efficiently, then $\mathsf{P}_1$ would not have been a computationally bounded algorithm.

## 1.4 Perspective of Public Key Cryptography

Over the years, cryptographic research has proposed solutions to many different cryptographic tasks under a growing number of (unproven) computational hardness assumptions. To some extent, this state of affairs is unavoidable, since the security of almost any cryptographic primitive implies the existence of one-way functions [Impagliazzo and Luby, 1989] (which in particular implies that $P \neq NP$). Yet, all various assumptions can essentially be divided into two main types: *private key* cryptography and *public key* cryptography [Impagliazzo, 1995]. The former type is better understood: A series of works have shown that the unstructured form of hardness guaranteed by one-way functions is sufficient to construct many complex and structured primitives such as pseudorandom generators [Håstad et al., 1999], pseudorandom functions [Goldreich et al., 1986] and permutations [Luby and Rackoff, 1988], commitment schemes [Naor, 1991, Haitner et al., 2009], universal one-way hash functions [Rompel, 1990], zero-knowledge proofs [Goldreich et al., 1987], and more. However, reductions are much less common outside the one-way functions regime, particularly when constructing public-key primitives. In the famous work of Impagliazzo and Rudich [1989] they gave the first evidence that *public key* cryptography assumptions are strictly stronger than one-way functions, by showing that key-agreement, which enables two parties to exchange secret messages over open channels, cannot be constructed from one-way functions in a black-box way.

Our work shows that a *balanced* adversary for the ADA game that has the structure of all known attacks, is a primitive that belongs to the public-key cryptography type. In particular, if public-key cryptography does not exist, it could be possible to construct a computationally bounded mechanism that can handle more than $\Theta(n^2)$ adaptive queries of a *balanced* adversary (i.e., we currently do not have a negative result that rules out this possibility).

## 1.5 Other Related Work

Nissim et al. [2018] presented a variant of the lower bound of Steinke and Ullman [2015a] that aims to reduce the number of queries used by the attacker. However, their resulting lower bound only holds for a certain family of mechanisms, and it does not rule out all computationally efficient mechanisms.

Dinur et al. [2023] revisited and generalized the lower bounds of Hardt and Ullman [2014] and Steinke and Ullman [2015a] by showing that they are a consequence of a space bottleneck rather than a sampling bottleneck. Yet, as in the works by Hardt, Steinke, and Ullman, the attack by Dinur et al. relies on the ability to choose the underlying distribution $\mathcal{D}$ and inject secret trapdoors in it, and hence it utilizes an *imbalanced* adversary.

Recently, lower bounds constructions for the ADA problem were used as a tool for constructing (conditional) lower bounds for other problems, such as the space complexity of *adaptive streaming algorithms* [Kaplan et al., 2021] and the time complexity of *dynamic algorithms* Beimel et al. [2022]. Our lower bound for the ADA problem is qualitatively stronger than previous lower bounds (as the adversary we construct has less power). Thus, our lower bound could potentially yield new connections and constructions in additional settings.

## 1.6 Conclusion and Open Problems

In this work we present the balanced adversarial model for the ADA problem, and show that the existence of a balanced adversary that has the structure of all previously known attacks is equivalent to the existence of public-key cryptography. Yet, we do not know what is the truth outside of the public-key cryptography world. Can we present a different type of efficient attack that is based on weaker hardness assumptions (like one-way functions)? Or is it possible to construct an efficient mechanism that answer more than $\Theta(n^2)$ adaptive queries assuming that public-key cryptography does not exist? We also leave open similar questions regarding the information theoretic case. We currently do not know whether it is possible to construct an unbounded mechanism that answers exponential number of queries for any distribution $\mathcal{D}$ (regardless of its domain size).

In a broader perspective, lower bounds such as ours show that no general solution exists for a problem. They often use unnatural inputs or distributions and rely on cryptographic assumptions. They are important as guidance for how to proceed with a problem, e.g., search for mechanisms that would succeed if the underlying distribution is from a "nice" family of distributions.

## 2 Preliminaries

### 2.1 Notations

We use calligraphic letters to denote sets and distributions, uppercase for random variables, and lowercase for values and functions. For $n \in \mathbb{N}$, let $[n] = \{1, 2, \ldots, n\}$. Let $\mathrm{neg}(n)$ stand for a negligible function in $n$, i.e., a function $\nu(n)$ such that for every constant $c > 0$ and large enough $n$ it holds that $\nu(n) < n^{-c}$. For $n \in \mathbb{N}$ we denote by $1^n$ the $n$-size string $1 \ldots 1$ ($n$ times). Let PPT stand for probabilistic polynomial time. We say that a pair of algorithms $\mathsf{A} = (\mathsf{A}_1, \mathsf{A}_2)$ is PPT if both $\mathsf{A}_1$ and $\mathsf{A}_2$ are PPT algorithms.

### 2.2 Distributions and Random Variables

Given a distribution $\mathcal{D}$, we write $x \sim \mathcal{D}$, meaning that $x$ is sampled according to $\mathcal{D}$. For a multiset $\mathcal{S}$, we denote by $\mathcal{U}_\mathcal{S}$ the uniform distribution over $\mathcal{S}$, and let $x \leftarrow \mathcal{S}$ denote that $x \sim \mathcal{U}_\mathcal{S}$. For a distribution $\mathcal{D}$ and a value $n \in \mathbb{N}$, we denote by $\mathcal{D}^n$ the distribution of $n$ i.i.d. samples from $\mathcal{D}$. For a distribution $\mathcal{D}$ over $\mathcal{X}$ and a query $q \colon \mathcal{X} \to [-1, 1]$, we abuse notation and denote $q(\mathcal{D}) := \mathrm{E}_{x \sim \mathcal{D}}[q(x)]$, and similarly for $\mathcal{S} = (x_1, \ldots, x_n) \in \mathcal{X}^*$ we abuse notation and denote $q(\mathcal{S}) := \mathrm{E}_{x \leftarrow \mathcal{S}}[q(x)] = \frac{1}{n} \sum_{i=1}^n x_i$.

### 2.3 Cryptographic Primitives

#### 2.3.1 Key Agreement Protocols

The most basic public-key cryptographic primitive is a (1-bit) *key-agreement* protocol, defined below.

**Definition 2.1** (key-agreement protocol)**.** *Let $\pi$ be a two party protocol between two* interactive PPT *algorithms* $\mathsf{P}_1$ *and* $\mathsf{P}_2$*, each outputs 1-bit. Let $\pi(1^n)$ denote a random execution of the protocol on joint input $1^n$ (the security parameter), and let $O_n^1, O_n^2$ and $T_n$ denote the random variables of* $\mathsf{P}_1$*'s output,* $\mathsf{P}_2$*'s output, and the transcript (respectively) in this execution. We say that $\pi$ is an $(\alpha, \beta)$-key-agreement protocol if the following holds for any* PPT *("eavesdropper")* $\mathsf{A}$ *and any $n \in \mathbb{N}$:*

**Agreement:** $\Pr\big[O_n^1 = O_n^2\big] \geq \alpha(n)$*, and*

**Secrecy:** $\Pr\big[\mathsf{A}(T_n) = O_n^1\big] \leq \beta(n)$*.*

*We say that $\pi$ is a* fully-secure key-agreement *protocol if it is an $(1 - \mathrm{neg}(n), 1/2 + \mathrm{neg}(n))$-key-agreement protocol.*

#### 2.3.2 Identity-Based Encryption

An Identity-Based Encryption (IBE) scheme [Shamir, 1984, Cocks, 2001, Boneh and Franklin, 2001] consists of four PPT algorithms $(\mathsf{Setup}, \mathsf{KeyGen}, \mathsf{Encrypt}, \mathsf{Decrypt})$ defined as follows:

$\mathsf{Setup}(1^\lambda)$: given the security parameter $\lambda$, it outputs a master public key $\mathsf{mpk}$ and a master secret key $\mathsf{msk}$.

$\mathsf{KeyGen}(\mathsf{msk}, \mathsf{id})$: given the master secret key $\mathsf{msk}$ and an identity $\mathsf{id} \in [n]$, it outputs a decryption key $\mathsf{sk}_{\mathsf{id}}$.

$\mathsf{Encrypt}(\mathsf{mpk}, \mathsf{id}, \mathsf{m})$: given the master public key $\mathsf{mpk}$, and identity $\mathsf{id} \in [n]$ and a message $\mathsf{m}$, it outputs a ciphertext $\mathsf{ct}$.

$\mathsf{Decrypt}(\mathsf{sk}_{\mathsf{id}}, \mathsf{ct})$: given a secret key $\mathsf{sk}_{\mathsf{id}}$ for identity $\mathsf{id}$ and a ciphertext $\mathsf{ct}$, it outputs a string $\mathsf{m}$.

The following are the properties of such an encryption scheme:

**Completeness:** For all security parameter $\lambda$, identity $\mathsf{id} \in [n]$ and a message $\mathsf{m}$, with probability $1$ over $(\mathsf{mpk}, \mathsf{msk}) \sim \mathsf{Setup}(1^\lambda)$ and $\mathsf{sk_{id}} \sim \mathsf{KeyGen}(\mathsf{msk}, \mathsf{id})$ it holds that

$$\mathsf{Decrypt}(\mathsf{sk_{id}}, \mathsf{Encrypt}(\mathsf{mpk}, \mathsf{id}, \mathsf{m})) = \mathsf{m}$$

**Security:** For any PPT adversary $\mathsf{A} = (\mathsf{A_1}, \mathsf{A_2})$ it holds that:

$$\Pr[IND_\mathsf{A}^{IBE}(1^\lambda) = 1] \le 1/2 + \mathsf{neg}(\lambda)$$

where $IND_\mathsf{A}^{IBE}$ is shown in Experiment 2.2.[5]

---

**Experiment 2.2** ($IND_\mathsf{A}^{IBE}(1^\lambda)$)**.**

1. $(\mathsf{mpk}, \mathsf{msk}) \sim \mathsf{Setup}(1^\lambda)$.
2. $(\mathsf{id}^*, (\mathsf{m}_1^0, \ldots, \mathsf{m}_k^0), (\mathsf{m}_1^1, \ldots, \mathsf{m}_k^1), \mathsf{st}) \sim \mathsf{A}_1^{\mathsf{KeyGen}(\mathsf{msk}, \cdot)}(\mathsf{mpk})$ *where* $\left|\mathsf{m}_i^0\right| = \left|\mathsf{m}_i^1\right|$ *for every* $i \in [k]$ *and for each query* $\mathsf{id}$ *by* $\mathsf{A}_1$ *to* $\mathsf{KeyGen}(\mathsf{msk}, \cdot)$ *we have that* $\mathsf{id} \ne \mathsf{id}^*$.
3. *Sample* $b \leftarrow \{0, 1\}$.
4. *Sample* $\mathsf{ct}_i^* \sim \mathsf{Encrypt}(\mathsf{mpk}, \mathsf{id}^*, \mathsf{m}_i^b)$ *for every* $i \in [k]$.
5. $b' \sim \mathsf{A}_2^{\mathsf{KeyGen}(\mathsf{msk}, \cdot)}(\mathsf{mpk}, (\mathsf{ct}_1^*, \ldots \mathsf{ct}_k^*), \mathsf{st})$ *where for each query* $\mathsf{id}$ *by* $\mathsf{A}_2$ *to* $\mathsf{KeyGen}(\mathsf{msk}, \cdot)$ *we have that* $\mathsf{id} \ne \mathsf{id}^*$.
6. *Output* $1$ *if* $b = b'$ *and* $0$ *otherwise.*

---

Namely, the adversary chooses two sequences of messages $(\mathsf{m}_1^0, \ldots, \mathsf{m}_k^0)$ and $(\mathsf{m}_1^1, \ldots, \mathsf{m}_k^1)$, and gets encryptions of either the first sequence or the second one, where the encryptions made for identity $\mathsf{id}^*$ that the adversary does not hold its key (not allowed to query $\mathsf{KeyGen}$ on input $\mathsf{id}^*$). The security requirement means that she cannot distinguish between the two cases (except with negligible probability).

**Theorem 2.3** (Döttling and Garg [2021])**.** *Assume that the Computational Diffie-Hellman (CDH) Problem is hard. Then there exists an IBE scheme* $\mathcal{E} = (\mathsf{Setup}, \mathsf{KeyGen}, \mathsf{Encrypt}, \mathsf{Decrypt})$ *for* $n$ *identities such that given a security parameter* $\lambda$, *the master keys and each decryption key are of size* $O(\lambda \cdot \log n)$.[6]

## 2.4 Balanced Adaptive Data Analysis

As described in the introduction, the mechanism plays a game with a *balanced* adversary that consists of two (isolated) algorithms: a *sampler* $\mathsf{A}_1$, which chooses a distribution $\mathcal{D}$ over a domain $\mathcal{X}$ and provides $n$ i.i.d. samples to $\mathsf{M}$, and an *analyst* $\mathsf{A}_2$, which asks the adaptive queries about the distribution. Let $\mathsf{ADA}_{n,\ell,\mathcal{X}}[\mathsf{M}, \mathsf{A} = (\mathsf{A}_1, \mathsf{A}_2)]$ denote Game 1.3 on public inputs $n$ - the number of samples, $\ell$ - the number of queries, and $\mathcal{X}$ - the domain. We denote by output $1$ the case that $\mathsf{M}$ fails in the game, and $0$ otherwise. Since this work mainly deals with computationally bounded algorithms that we would like to model as PPT algorithms, we provide $n$ and $\ell$ in unary representation. We also assume for simplicity that $\mathcal{X}$ is finite, which allows to represent each element as a binary vector of dimension $\lceil \log |\mathcal{X}| \rceil$, and we provide the dimension in unary representation as well.

All previous negative results (Hardt and Ullman [2014], Steinke and Ullman [2015a], and Dinur et al. [2023]) where achieved by reduction to a restricted family of mechanisms, called *natural* mechanisms (Definition 1.8). These are algorithms that can only evaluate the query on the sample points they are given.

For *natural* mechanisms (even unbounded ones), the following was proven.

**Theorem 2.4** (Hardt and Ullman [2014], Steinke and Ullman [2015a])**.** *There exists a pair of* PPT *algorithms* $\widetilde{\mathsf{A}} = (\widetilde{\mathsf{A}}_1, \widetilde{\mathsf{A}}_2)$ *such that for every* natural *mechanism* $\widetilde{\mathsf{M}}$ *and every large enough* $n$ *and* $\ell = \Theta(n^2)$ *it holds that*

$$\Pr\left[\mathsf{ADA}_{n,\ell,\mathcal{X}=[2000n]}[\widetilde{\mathsf{M}}, \widetilde{\mathsf{A}}] = 1\right] > 3/4. \tag{1}$$

---

[5]The IBE security experiment is usually described as Experiment 2.2 with $k = 1$ (i.e., encrypting a single message). Yet, it can be extended to any sequence of messages using a simple reduction.

[6]The construction can also be based on the hardness of *factoring*.

*In particular, $\widetilde{\mathsf{A}}_1$ always chooses the uniform distribution over $[2000n]$, and $\widetilde{\mathsf{A}}_2$ uses only queries over the range $\{-1, 0, 1\}$.*

# 3  Constructing a Balanced Adversary via IBE

We prove that if an IBE scheme exists, then there is an efficient reduction to *natural* mechanisms that holds against any PPT mechanism, yielding a general lower bound for the computational case.

**Theorem 3.1** (Restatement of Theorem 1.5)**.** *Assume the existence of an IBE scheme $\mathcal{E}$ that supports $m = 2000n$ identities with security parameter $\lambda = \lambda(n)$ s.t. $n \leq \mathrm{poly}(\lambda)$ (e.g., $\lambda = n^{0.1}$) using keys of length $k = k(n)$. Then there exists a PPT balanced adversary $\mathsf{A} = (\mathsf{A}_1, \mathsf{A}_2)$ and $\ell = \Theta(n^2) + k$ such that for every PPT mechanism $\mathsf{M}$ it holds that*

$$\Pr\Big[\mathsf{ADA}_{n,\ell,\mathcal{X}=[m]\times\{0,1\}^{2k}}[\mathsf{M}, \mathsf{A}] = 1\Big] > 3/4 - \mathrm{neg}(n).$$

The proof of the theorem is given in the supplementary material. Note that we use a domain $\mathcal{X}$ with $\log|\mathcal{X}| = 2k + \log n + O(1)$, and by applying Theorem 2.3, the lower bound holds for $k = O(\lambda \cdot \log n)$ under the CDH hardness assumption.

# 4  Reduction to Natural Mechanisms Implies Key Agreement

We prove that any PPT *balanced* adversary $\mathsf{A} = (\mathsf{A}_1, \mathsf{A}_2)$ that has the structure of all known lower bounds (Hardt and Ullman [2014], Steinke and Ullman [2015a], Dinur et al. [2023] and ours in Section 3), can be used to construct a *key-agreement* protocol.

All known constructions use an adversary $\mathsf{A}$ that wraps the adversary $\widetilde{\mathsf{A}}$ for the natural mechanisms case (Theorem 2.4) by forcing every mechanism $\mathsf{M}$ to behave *naturally* using cryptography. In particular, they all satisfy Properties 1 and 2 from Section 1.3.2.

The formal statement is given in the following theorem. The proof is provided in the supplementary material.

**Theorem 4.1** (Restatement of Theorem 1.6)**.** *Assume the existence of a PPT adversary $\mathsf{A} = (\mathsf{A}_1, \mathsf{A}_2)$ and functions $\ell = \ell(n) \leq \mathrm{poly}(n)$ and $\mathcal{X} = \mathcal{X}(n)$ with $\log|\mathcal{X}| \leq \mathrm{poly}(n)$ such that the following holds: Let $n \in \mathbb{N}$ and consider a random execution of $\mathsf{ADA}_{n,\ell,\mathcal{X}}[\mathsf{M}, \mathsf{A}]$ where $\mathsf{M}$ is the mechanism that given a sample $\mathcal{S}$ and a query $q$, answers the empirical mean $q(\mathcal{S})$. Let $D_n$ and $Q_n$ be the (r.v.'s of the) values of $\mathcal{D}$ and $q = q_\ell$ (the last query) in the execution (respectively), let $T_n$ be the transcript of the execution between the analyst $\mathsf{A}_2$ and the mechanism $\mathsf{M}$ (i.e., the queries and answers), and let $V_n$ be the view of $\mathsf{A}_2$ at the end of the execution (without loss of generality, its input, random coins and the transcript). Assume that*

1. *$\exists$ PPT algorithm $\mathsf{F}$ s.t. $\forall n \in \mathbb{N}: \ \Pr\big[|\mathsf{F}(V_n) - Q_n(D_n)| \leq n^{-1/10}\big] \geq 1 - \mathrm{neg}(n)$, and*

2. *$\forall$ PPT algorithm $\mathsf{G}$ and $\forall n \in \mathbb{N}: \ \Pr[|\mathsf{G}(T_n) - Q_n(D_n)| \leq 1/10] \leq 1/4 + \mathrm{neg}(n)$.*

*Then using $\mathsf{A}$ and $\mathsf{F}$ it is possible to construct a fully-secure key-agreement protocol.*

Note that Assumption 1 in Theorem 4.1 formalizes the first property in which the analyst knows a good estimation of the true answer, and the PPT algorithm $\mathsf{F}$ is the assumed knowledge extractor. Assumption 2 in Theorem 4.1 formalizes the second property which states that the mechanism, which answers the empirical mean along the way, will fail in the last query, no matter how it chooses to act (this behavior is captured with the PPT algorithm $\mathsf{G}$), and moreover, it is enough to assume that this requirement only holds with respect to to transcript of the execution, and not with respect to the view of the mechanism.

We refer to the full version of the paper, given in the supplementary material, for all the missing proofs.

## Acknowledgments and Disclosure of Funding

Kobbi Nissim is partially supported by NSF grant No. CNS-2001041 and a gift to Georgetown University. Uri Stemmer is partially supported by the Israel Science Foundation (grant 1871/19) and by Len Blavatnik and the Blavatnik Family foundation. Eliad Tsfadia is partially supported by the Fulbright Program and a gift to Georgetown University.

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
