# Adaptive Data Analysis in a Balanced Adversarial Model

Kobbi Nissim[*]    Uri Stemmer[†]    Eliad Tsfadia[‡]

October 26, 2023

## Abstract

In adaptive data analysis, a mechanism gets $n$ i.i.d. samples from an unknown distribution $\mathcal{D}$, and is required to provide accurate estimations to a sequence of adaptively chosen statistical queries with respect to $\mathcal{D}$. Hardt and Ullman [HU14] and Steinke and Ullman [SU15b] showed that, in general, it is computationally hard to answer more than $\Theta(n^2)$ adaptive queries, assuming the existence of one-way functions.

However, these negative results strongly rely on an adversarial model that significantly advantages the adversarial analyst over the mechanism, as the analyst, who chooses the adaptive queries, also chooses the underlying distribution $\mathcal{D}$. This imbalance raises questions with respect to the applicability of the obtained hardness results – an analyst who has complete knowledge of the underlying distribution $\mathcal{D}$ would have little need, if at all, to issue statistical queries to a mechanism which only holds a finite number of samples from $\mathcal{D}$.

We consider more restricted adversaries, called *balanced*, where each such adversary consists of two separate algorithms: The *sampler* who is the entity that chooses the distribution and provides the samples to the mechanism, and the *analyst* who chooses the adaptive queries, but has no prior knowledge of the underlying distribution (and hence has no a priori advantage with respect to the mechanism).

We improve the quality of previous lower bounds by revisiting them using an efficient *balanced* adversary, under standard public-key cryptography assumptions. We show that these stronger hardness assumptions are unavoidable in the sense that any computationally bounded *balanced* adversary that has the structure of all known attacks, implies the existence of public-key cryptography.

---

[*]Department of Computer Science, Georgetown University. E-mail: `kobbi.nissim@georgetown.edu`. Work partially supported by NSF grant No. CNS-2001041 and a gift to Georgetown University.

[†]Blavatnik School of Computer Science, Tel Aviv University, and Google Research. E-mail: `u@uri.co.il`. Work partially supported by the Israel Science Foundation (grant 1871/19) and by Len Blavatnik and the Blavatnik Family foundation.

[‡]Department of Computer Science, Georgetown University. E-mail: `eliadtsfadia@gmail.com`. Work supported in part by the Fulbright Program and a gift to Georgetown University.

# Contents

# 1 Introduction

Statistical validity is a widely recognized crucial feature of modern science. Lack of validity – popularly known as the *replication crisis* in science poses a serious threat to the scientific process and also to the public's trust in scientific findings.

One of the factors leading to the replication crisis is the inherent adaptivity in the data analysis process. To illustrate adaptivity and its effect, consider a data analyst who is testing a specific research hypothesis. The analyst gathers data, evaluates the hypothesis empirically, and often finds that their hypothesis is not supported by the data, leading to the formulation and testing of more hypotheses. If these hypotheses are tested and formed based on the same data (as acquiring fresh data is often expensive or even impossible), then the process is of *adaptive data analysis* (ADA) because the choice of hypotheses depends on the data. However, ADA no longer aligns with classical statistical theory, which assumes that hypotheses are selected independently of the data (and preferably before gathering data). ADA may lead to overfitting and hence false discoveries.

Statistical validity under ADA is a fundamental problem in statistics, that has received only partial answers. A recent line of work, initiated by [DFH+15c] and includes [HU14; DFH+15a; DFH+15b; SU15b; SU15a; BNS+16; RRST16; RZ16; Smi17; FS17; NSS+18; FS18; SL19; JLN+20; FRR20; DK22; KSS22; DSWZ23; Bla23]

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

**Fact 2.1** (Hoeffding's inequality). *Let $X_1, \ldots, X_n$ be i.i.d. random variables over $[-1, 1]$ with expectation $\mu$. Then*

$$\Pr\left[\left|\frac{1}{n} \sum_{i=1}^n X_i - \mu\right| \geq \alpha\right] \leq 2 \cdot e^{-\alpha^2 n / 2}$$

### 2.3 Cryptographic Primitives

#### 2.3.1 Key Agreement Protocols

The most basic public-key cryptographic primitive is a (1-bit) *key agreement* protocol, defined below.

**Definition 2.2** (key-agreement protocol). *Let $\pi$ be a two party protocol between two* interactive *PPT algorithms $\mathsf{P}_1$ and $\mathsf{P}_2$, each outputs 1-bit. Let $\pi(1^n)$ denote a random execution of the protocol on joint input $1^n$ (the security parameter), and let $O_n^1, O_n^2$ and $T_n$ denote the random variables of $\mathsf{P}_1$'s output, $\mathsf{P}_2$'s output, and the transcript (respectively) in this execution. We say that $\pi$ is an $(\alpha, \beta)$-key-agreement protocol if the following holds for any PPT (i.e., "eavesdropper") $\mathsf{A}$ and every $n \in \mathbb{N}$:*

**Agreement:** $\Pr\left[O_n^1 = O_n^2\right] \geq \alpha(n)$, *and*

**Secrecy:** $\Pr\left[\mathsf{A}(T_n) = O_n^1\right] \leq \beta(n)$.

*We say that $\pi$ is a* fully-secure key-agreement *protocol if it is an $(1 - \mathrm{neg}(n), 1/2 + \mathrm{neg}(n))$-key-agreement protocol.*

We use the following weaker type of agreement.

**Definition 2.3** (Approximate agreement protocol)**.** *Let $\pi$ be a two party protocol between two interactive* PPT *algorithms* $\mathsf{P}_1$ *and* $\mathsf{P}_2$, *each outputs a value in* $[-1, 1]$, *and denote by* $O_n^1, O_n^2 \in [-1, 1]$ *and* $T_n$ *the random variables of the outputs of* $\mathsf{P}_1$, $\mathsf{P}_2$ *and transcript (respectively) in a random the execution* $\pi(1^n)$. *We say that* $\pi$ *is an* $(\alpha, \beta)$-*approximate agreement protocol if the following holds for any* PPT $\mathsf{A}$ *and* $n \in \mathbb{N}$:

**Approximate Agreement:** $\Pr\big[\big|O_n^1 - O_n^2\big| \leq \alpha(n)\big] \geq 1 - \mathrm{neg}(n)$, *and*

**Secrecy:** $\Pr\big[\big|\mathsf{A}(T_n) - O_n^1\big| \leq \beta(n)\big] \leq 1 - n^{-\Omega(1)}$.

Namely, when $\alpha(n) < \beta(n)$, the parties in an *approximate agreement* protocol do not agree on the same value, but are able to output values that are closer to each other than any prediction of a PPT "eavesdropper" adversary. We show that such approximate agreement suffices for constructing a fully-secure key-agreement.

**Theorem 2.4.** *Let* $\alpha, \beta \colon \mathbb{N} \to [0, 1]$ *be efficiently computable functions such that* $\alpha(n)/\beta(n) \leq n^{-\Omega(1)}$ *and* $\alpha(n) \cdot \beta(n) \geq 2^{-n}$ *for large enough* $n$. *If there exists an* $(\alpha, \beta)$-*approximate-agreement protocol, then there exists a fully-secure key-agreement protocol.*

A close variant of Theorem 2.4 is implicitly proved in [HMST22] (with better parameters, but in a more complicated setting). For completeness, we give a full proof of Theorem 2.4 in Appendix A.

### 2.3.2 Identity-Based Encryption

An Identity-Based Encryption (IBE) scheme [Sha84; Coc01; BF01] consists of four PPT algorithms (Setup, KeyGen, Encrypt, Decrypt) defined as follows:

- Setup($1^\lambda$): given the security parameter $\lambda$, it outputs a master public key mpk and a master secret key msk.

- KeyGen(msk, id): given the master secret key msk and an identity id $\in [n]$, it outputs a decryption key $\mathsf{sk}_\mathsf{id}$.

- Encrypt(mpk, id, m): given the master public key mpk, and identity id $\in [n]$ and a message m, it outputs a ciphertext ct.

- Decrypt($\mathsf{sk}_\mathsf{id}$, ct): given a secret key $\mathsf{sk}_\mathsf{id}$ for identity id and a ciphertext ct, it outputs a string m.

The following are the properties of such an encryption scheme:

- **Completeness:** For all security parameter $\lambda$, identity id $\in [n]$ and a message m, with probability 1 over (mpk, msk) $\sim$ Setup($1^\lambda$) and $\mathsf{sk}_\mathsf{id} \sim$ KeyGen(msk, id) it holds that

$$\mathsf{Decrypt}(\mathsf{sk}_\mathsf{id}, \mathsf{Encrypt}(\mathsf{mpk}, \mathsf{id}, \mathsf{m})) = \mathsf{m}$$

- **Security:** For any PPT adversary $A = (A_1, A_2)$ it holds that:

$$\Pr[IND_A^{IBE}(1^\lambda) = 1] \leq 1/2 + \text{neg}(\lambda)$$

where $IND_A^{IBE}$ is shown in Experiment 2.5.[5]

---

**Experiment 2.5** ($IND_A^{IBE}(1^\lambda)$).

1. $(\mathsf{mpk}, \mathsf{msk}) \sim \mathsf{Setup}(1^\lambda)$.

2. $(\mathsf{id}^*, (\mathsf{m}_1^0, \ldots, \mathsf{m}_k^0), (\mathsf{m}_1^1, \ldots, \mathsf{m}_k^1), \mathsf{st}) \sim A_1^{\mathsf{KeyGen}(\mathsf{msk}, \cdot)}(\mathsf{mpk})$ *where* $\left|\mathsf{m}_i^0\right| = \left|\mathsf{m}_i^1\right|$ *for every* $i \in [k]$ *and for each query* $\mathsf{id}$ *by* $A_1$ *to* $\mathsf{KeyGen}(\mathsf{msk}, \cdot)$ *we have that* $\mathsf{id} \neq \mathsf{id}^*$.

3. *Sample* $b \leftarrow \{0, 1\}$.

4. *Sample* $\mathsf{ct}_i^* \sim \mathsf{Encrypt}(\mathsf{mpk}, \mathsf{id}^*, \mathsf{m}_i^b)$ *for every* $i \in [k]$.

5. $b' \sim A_2^{\mathsf{KeyGen}(\mathsf{msk}, \cdot)}(\mathsf{mpk}, (\mathsf{ct}_1^*, \ldots \mathsf{ct}_k^*), \mathsf{st})$ *where for each query* $\mathsf{id}$ *by* $A_2$ *to* $\mathsf{KeyGen}(\mathsf{msk}, \cdot)$ *we have that* $\mathsf{id} \neq \mathsf{id}^*$.

6. *Output* 1 *if* $b = b'$ *and* 0 *otherwise.*

---

Namely, the adversary chooses two sequences of messages $(\mathsf{m}_1^0, \ldots, \mathsf{m}_k^0)$ and $(\mathsf{m}_1^1, \ldots, \mathsf{m}_k^1)$, and gets encryptions of either the first sequence or the second one, where the encryptions made for identity $\mathsf{id}^*$ that the adversary does not hold its key (not allowed to query $\mathsf{KeyGen}$ on input $\mathsf{id}^*$). The security requirement means that she cannot distinguish between the two cases (except with negligible probability).

Shamir [Sha84] was the first to consider the problem of constructing an IBE scheme that can support many identities using small keys. The first IBE schemes were realized by Boneh and Franklin [BF01] and Cocks [Coc01], but their security analyses were based on non-standard cryptographic assumptions: the quadratic residuocity assumption [Coc01] and assumptions on groups with billinear map [BF01]. More recently, Döttling and Garg [DG21] and Blazy and Kakvi [BK22] have managed to construct an IBE scheme based on the standard Computational Diffie-Hellman (CDH) hardness assumption. Below we summarize the construction properties of [DG21].

**Theorem 2.6** ([DG21]). *Assume that the Computational Diffie-Hellman (CDH) Problem is hard. Then there exists an IBE scheme* $\mathcal{E} = (\mathsf{Setup}, \mathsf{KeyGen}, \mathsf{Encrypt}, \mathsf{Decrypt})$ *for* $n$ *identities such that given a security parameter* $\lambda$, *the master keys and each decryption key are of size* $O(\lambda \cdot \log n)$.[6]

## 2.4 Balanced Adaptive Data Analysis

Adaptive data analysis is modeled as a game between a *mechanism* $M$ and an *analyst* $A$. The mechanism gets as input $n$ i.i.d. samples $x_1, \ldots, x_n$ from an (unknown) distribution $\mathcal{D}$ over a domain $\mathcal{X}$, and its goal is to answer statistical queries about $\mathcal{D}$, produced by the analyst. Namely, when $A$ sends a statistical query $q \colon \mathcal{X} \to [-1, 1]$, $M$ is required to return an answer $y \in [-1, 1]$ that is close to $q(\mathcal{D}) = \mathrm{E}_{x \sim \mathcal{D}}[q(x)]$.

---

[5]The IBE security experiment is usually described as Experiment 2.5 with $k = 1$ (i.e., encrypting a single message). Yet, it can be extended to any sequence of messages using a simple reduction.

[6]The construction can also be based on the hardness of *factoring*.

In this work we investigate a *balanced* setting where the adversarial analyst does not have an informational advantage over the mechanism. Namely, the analyst has no knowledge about the underline distribution which the mechanism does not have. We model this situation by separating between the analyst from the underline distribution, as follows:

The mechanism plays a game with a *balanced* adversary that consists of two (isolated) algorithms: a *sampler* $A_1$, which chooses a distribution $\mathcal{D}$ over a domain $\mathcal{X}$ and provides $n$ i.i.d. samples to M, and an *analyst* $A_2$, which asks the adaptive queries about the distribution (see Game 2.7). The public inputs for the ADA game are the number of samples $n$, the number of queries $\ell$ and the domain $\mathcal{X}$. Since this work mainly deals with computationally bounded algorithms that we would like to model as PPT algorithms, we provide $n$ and $\ell$ in unary representation. We also assume for simplicity that $\mathcal{X}$ is finite, which allows to represent each element as a binary vector of dimension $\lceil \log|\mathcal{X}| \rceil$, and we provide the dimension in unary representation as well.

---

**Game 2.7** ($\mathrm{ADA}_{n,\ell,\mathcal{X}}[\mathsf{M}, \mathsf{A} = (\mathsf{A}_1, \mathsf{A}_2)]$, redefinition of Game 1.3)**.**

*Public inputs: Number of samples $1^n$, number of queries $1^\ell$, and a domain $\mathcal{X}$ (represented as $1^{\lceil \log|\mathcal{X}| \rceil}$).*

*Operation:*

1. $\mathsf{A}_1$ *chooses a distribution $\mathcal{D}$ over $\mathcal{X}$ (specified by a sampling algorithm) and sends $\mathcal{S} = (x_1, \ldots, x_n) \sim \mathcal{D}^n$ to M (i.e., applies the sampling algorithm $n$ times).*

2. *For $i = 1, \ldots, \ell$:*

    (a) $\mathsf{A}_2$ *sends a query $q_i \colon \mathcal{X} \mapsto [-1, 1]$ to M.*

    (b) M *sends an answer $y_i \in [-1, 1]$ to $\mathsf{A}_2$.*
    *(As $\mathsf{A}_2$ and M are stateful, $q_i$ and $y_i$ may depend on the previous messages.)*

3. *The outcome is one if $\exists i \in [\ell]$ s.t. $|y_i - q_i(\mathcal{D})| > 1/10$, and zero otherwise.*

---

All previous negative results ([HU14; SU15b; DSWZ23]) where achieved by reduction to a restricted family of mechanisms, called *natural* mechanisms (Definition 1.8). These are algorithms that can only evaluate the query on the sample points they are given.

For *natural* mechanisms (even unbounded ones), the following was proven.

**Theorem 2.8** ([HU14; SU15b])**.** *There exists a pair of PPT algorithms $\widetilde{\mathsf{A}} = (\widetilde{\mathsf{A}}_1, \widetilde{\mathsf{A}}_2)$ such that for every* natural *mechanism $\widetilde{\mathsf{M}}$ and every large enough $n$ and $\ell = \Theta(n^2)$ it holds that*

$$\Pr\left[\mathrm{ADA}_{n,\ell,\mathcal{X}=[2000n]}[\widetilde{\mathsf{M}}, \widetilde{\mathsf{A}}] = 1\right] > 3/4. \tag{1}$$

*In particular, $\widetilde{\mathsf{A}}_1$ always chooses the uniform distribution over $[2000n]$, and $\widetilde{\mathsf{A}}_2$ uses only queries over the range $\{-1, 0, 1\}$.*

The adversary $\widetilde{\mathsf{A}}$ from Theorem 2.8 uses queries that are based on random *interactive fingerprinting code* [SU15b] which enables to reconstruct most of the $n$ samples $\mathcal{S}$ given $\Theta(n^2)$ accurate answers that are only a function of the samples (as a *natural* mechanism must behave). Once

the samples are revealed to the analyst, it then prepares a last query that cannot be answered accurately by a *natural* mechanism (e.g., a query $q$ with $q(x) = 0$ for $x \in \mathcal{S}$, but with different values for elements $x \in \mathcal{X} \setminus \mathcal{S}$). In particular, this holds for the mechanism $\widetilde{\mathsf{M}}$ which given a query $q_i$ for $i \in [\ell - 1]$ and a sample $\mathcal{S}$, answers the empirical mean $q(\mathcal{S}) = \frac{1}{n} \sum_{x \in \mathcal{S}} x$. See the observation below which is used in Section 4.

**Observation 2.9** (Implicit in [SU15b])**.** *Let $\widetilde{\mathsf{M}}$ be a* natural *mechanism that given a sample $\mathcal{S} = (x_1, \ldots, x_n) \in \mathcal{X}^n$ and a query $q \colon \mathcal{X} \to [-1, 1]$ which is not the last one, answers the empirical mean $q(\mathcal{S}) = \frac{1}{n} \sum_{i=1}^{n} x_i$. Then in a random execution of $\mathsf{ADA}_{n,\ell,\mathcal{X}}[\widetilde{\mathsf{M}}, \widetilde{\mathsf{A}}]$ ($\widetilde{\mathsf{A}}, n, \ell, \mathcal{X}$ as in Theorem 2.8), $\widetilde{\mathsf{M}}$ will fail to answer the last query accurately, regardless of what natural strategy it uses for this query.*

## 3 Constructing a Balanced Adversary via IBE

In this section we prove that, under standard public-key cryptography assumptions (in particular, the existence of an IBE scheme), there is an efficient reduction to *natural* mechanisms that holds against any PPT mechanism, yielding a general lower bound for the computational case. In particular, we show that there exists a pair of PPT algorithms $\mathsf{A} = (\mathsf{A}_1, \mathsf{A}_2)$ such that for every PPT mechanism $\mathsf{M}$ it holds that

$$\Pr\left[\mathsf{ADA}_{n,\ell=\Theta(n^2),\mathcal{X}=\{0,1\}^{\tilde{O}(\lambda)}}[\mathsf{M}, \mathsf{A}] = 1\right] \geq 3/4 - \operatorname{neg}(n),$$

for any function $\lambda = \lambda(n)$ such that $n \leq \operatorname{poly}(\lambda)$ (e.g., $\lambda = n^{0.1}$). More specifically, we use a domain $\mathcal{X}$ with $\log|\mathcal{X}| = 2k + \log n + O(1)$, where $k = k(n)$ is the keys' length in the IBE scheme with security parameter $\lambda$ that supports $O(n)$ identities (by Theorem 2.6, such an IBE scheme exists with $k = O(\lambda \cdot \log n)$ under the CDH hardness assumption). $\mathsf{A}_1$ is defined in Algorithm 3.1, and $\mathsf{A}_2$ is defined in Algorithm 3.2.

---

**Algorithm 3.1** (Sampler $\mathsf{A}_1$)**.**

**Inputs:** *Number of samples $1^n$, number of queries $1^\ell$ and domain $\mathcal{X}$ (defined below). Let $m = 2000n$.*

**Oracle Access:** *$\mathsf{A}_1$ has access to an IBE scheme $\mathcal{E} = (\mathsf{Setup}, \mathsf{KeyGen}, \mathsf{Encrypt}, \mathsf{Decrypt})$ that supports $m$ identities with security parameter $\lambda = \lambda(n)$. Let $k = k(n)$ be the sizes of the keys in this scheme. Let $\mathcal{X} = [m] \times \{0,1\}^{2k}$.*

**Setting:** *$\mathsf{A}_1$ is the* sampler *in the $\mathsf{ADA}_{n,\ell,\mathcal{X}}$ game (Game 2.7) which provides to $\mathsf{M}$ $n$ i.i.d. samples from some underline distribution $\mathcal{D}$.*

**Operation:** *% Step 1 of Game 2.7:*

- *Sample $(\mathsf{mpk}, \mathsf{msk}) \sim \mathsf{Setup}(1^\lambda)$ and $\mathsf{sk}_j \sim \mathsf{KeyGen}(\mathsf{msk}, j)$ for every $j \in [m]$, and let $\mathcal{T} = \{(j, \mathsf{mpk}, \mathsf{sk}_j)\}_{j=1}^{m}$ and $\mathcal{D} = \mathcal{U}_\mathcal{T}$ (i.e., the uniform distribution over the triplets in $\mathcal{T}$).*

- *Send to $\mathsf{M}$ $n$ i.i.d. samples $\mathcal{S} \sim \mathcal{D}^n$.*

---

**Algorithm 3.2** (Analyst $A_2$).

**Inputs:** *Number of samples* $1^n$, *number of queries* $1^\ell$ *and a domain* $\mathcal{X}$ *(defined below). Let* $m = 2000n$.

**Oracle access:** $A_2$ *has access to an IBE scheme* $\mathcal{E} = (\mathsf{Setup}, \mathsf{KeyGen}, \mathsf{Encrypt}, \mathsf{Decrypt})$ *that supports* $m$ *identities with security parameter* $\lambda = \lambda(n)$. *Let* $k = k(n)$ *be the sizes of the keys in this scheme. Let* $\mathcal{X} = [m] \times \{0,1\}^{2k}$.

**Setting:** $A_2$ *is the* analyst *in the* $\mathsf{ADA}_{n,\ell,\mathcal{X}}$ *game (Game 2.7). It has access to the analyst* $\widetilde{A}_2$ *from Theorem 2.8 and it interacts with a (general, not necessarily natural) mechanism* $M$ *in* $\mathsf{ADA}_{n,\ell,\mathcal{X}}[M, (A_1, \cdot)]$ *where* $A_1$ *is Algorithm 3.1.*

*Operation:*

1. % *The first* $k$ *iterations in Step 2 of Game 2.7:*
   *For* $i = 1, 2, \ldots, k$:

   (a) % *Step 2a: Send to* $M$ *a query* $q_i$ *that on input* $(j, x, y) \in [m] \times \{0,1\}^k \times \{0,1\}^k$ *outputs* $x_i$.

   (b) % *Step 2b: Receive an answer from* $M$ *and round it for reconstructing the* $i^{\text{th}}$ *bit of* $\mathsf{mpk}$.

2. *Initialize an emulation of* $\widetilde{A}_2$ *in the game* $\mathsf{ADA}_{n,\ell-k,[m]}$.

3. % *The last* $\ell - k$ *iterations in Step 2 of Game 2.7:*
   *For* $i = 1, \ldots, \ell - k$:

   (a) *Obtain the* $i^{\text{th}}$ *query* $\tilde{q}_i \colon [m] \to \{-1, 0, 1\}$ *of the emulated* $\widetilde{A}_2$.

   (b) *For* $j \in [m]$, *compute* $\mathsf{ct}_{i,j} = \mathsf{Encrypt}(\mathsf{mpk}, j, \tilde{q}_i(j))$ *(i.e., encrypt* $\tilde{q}_i(j)$ *for identity* $j$).

   (c) *Define the query* $q_{i+k} \colon \mathcal{X} \to \{-1, 0, 1\}$ *that on input* $(j, x, y) \in [m] \times \{0,1\}^k \times \{0,1\}^k$ *outputs* $\mathsf{Decrypt}(y, \mathsf{ct}_{i,j})$. *The description of* $q_{i+k}$ *consists of* $\{\mathsf{ct}_{i,j}\}_{j \in [m]}$.

   (d) % *Step 2a: Send (the description of)* $q_{i+k}$ *to* $M$.

   (e) % *Step 2b: Receive an answer* $y_{i+k}$ *from* $M$.

   (f) *Send* $\tilde{y}_i = y_{i+k}$ *to the emulated* $\widetilde{A}_2$ *(as an answer to* $\tilde{q}_i$).

**Theorem 3.3** (Restatement of Theorem 1.5). *Assume the existence of an IBE scheme* $\mathcal{E}$ *that supports* $m = 2000n$ *identities with security parameter* $\lambda = \lambda(n)$ *s.t.* $n \le \text{poly}(\lambda)$ *using keys of length* $k = k(n)$. *Let* $A = (A_1, A_2)$, *where* $A_1$ *is Algorithm 3.1 and* $A_2$ *is Algorithm 3.2. Then there exists* $\ell = \Theta(n^2) + k$ *such that for every* PPT *mechanism* $M$ *it holds that*

$$\Pr\left[\mathsf{ADA}_{n,\ell,\mathcal{X} = [m] \times \{0,1\}^{2k}}[M, A] = 1\right] > 3/4 - \text{neg}(n).$$

*Proof.* Fix a PPT mechanism $M$ and large enough $n$. Consider the mechanism $\widetilde{M}$ defined in Algorithm 3.4 with respect to $M$. First, note that $\widetilde{M}$ is indeed *natural* since, upon receiving the query

$\tilde{q}_i$, it does not use the values $\{\tilde{q}_i(j)\}_{j\in[m]\setminus\mathcal{J}}$. Therefore, by Theorem 2.8 it holds that

$$\Pr\left[\mathsf{ADA}_{n,\ell-k,[m]}[\widetilde{\mathsf{M}},\widetilde{\mathsf{A}}]=1\right] \geq 3/4.$$

In the following, let $\widetilde{\mathsf{M}}'$ be an (unnatural) variant of $\widetilde{\mathsf{M}}$ that operates almost the same, except that in Step 4b, rather than sampling $\mathsf{ct}_{i,j} \sim \mathsf{Encrypt}(\mathsf{mpk}, j, 0)$ for $j \notin \mathcal{J}$, it samples $\mathsf{ct}_{i,j} \sim \mathsf{Encrypt}(\mathsf{mpk}, j, \tilde{q}_i(j))$. Note that both $\widetilde{\mathsf{M}}$ and $\widetilde{\mathsf{M}}'$, when playing in $\mathsf{ADA}_{n,\ell-k,[m]}[\cdot,\widetilde{\mathsf{A}}]$, emulate an execution of $\mathsf{ADA}_{n,\ell,\mathcal{X}}[\mathsf{M},\mathsf{A}]$, where the only difference between them is the values of $\mathsf{ct}_{i,j}$ for $j \notin \mathcal{J}$ that they send to the emulated $\mathsf{M}$ in each iteration $i$. But $\mathsf{M}$ is a $\mathrm{poly}(\lambda)$-time mechanism and its view in the emultations does not contain the keys $\{\mathsf{sk}_j\}_{j\notin\mathcal{J}}$. Therefore, by the security guarantee of the IBE scheme, the behavior of the emulated $\mathsf{M}$ is indistinguishable in both executions, yielding that

$$\Pr\left[\mathsf{ADA}_{n,\ell-k,[m]}[\widetilde{\mathsf{M}}',\widetilde{\mathsf{A}}]=1\right] \geq \Pr\left[\mathsf{ADA}_{n,\ell-k,[m]}[\widetilde{\mathsf{M}},\widetilde{\mathsf{A}}]=1\right] - \mathrm{neg}(n)$$
$$\geq 3/4 - \mathrm{neg}(n).$$

In the following we focus on proving that

$$\Pr\left[\mathsf{ADA}_{n,\ell,\mathcal{X}}[\mathsf{M},\mathsf{A}]=1\right] \geq \Pr\left[\mathsf{ADA}_{n,\ell-k,[m]}[\widetilde{\mathsf{M}}',\widetilde{\mathsf{A}}]=1\right], \tag{2}$$

which concludes the proof.

Let $T$, $Q_1, Y_1, \ldots, Q_\ell, Y_\ell$ be the (r.v.'s of the) values of $\mathcal{T}$, $q_1, y_1, \ldots, q_\ell, y_\ell$ (respectively) induced by $\mathsf{A}_2$ in a random execution of $\mathsf{ADA}_{n,\ell,\mathcal{X}}[\mathsf{M},\mathsf{A}]$, and let $E$ be the event that $\forall i \in [k] :$ $|Q_i(\mathcal{U}_T) - Y_i| \leq 1/10$. Similarly, let $T'$, $Q_1', Y_1', \ldots, Q_\ell', Y_\ell', \widetilde{Q}_1, \widetilde{Y}_1, \ldots, \widetilde{Q}_{\ell-k}, \widetilde{Y}_{\ell-k}$ be the (r.v.'s of the) values of $\mathcal{T}$, $q_1, y_1, \ldots, q_\ell, y_\ell, \tilde{q}_1, \tilde{y}_1, \ldots, \tilde{q}_{\ell-k}, \tilde{y}_{\ell-k}$ induced by $\widetilde{\mathsf{M}}'$ in an (independent) execution of $\mathsf{ADA}_{n,\ell-k,[m]}[\widetilde{\mathsf{M}}',\widetilde{\mathsf{A}}]$, and let $E'$ be the event that $\forall i \in [k] : |Q_i'(\mathcal{U}_{T'}) - Y_i'| \leq 1/10$. By construction, the following holds:

1. $\Pr[E] = \Pr[E']$ (holds since $\widetilde{\mathsf{M}}'$, as $\widetilde{\mathsf{M}}$, perfectly emulates the first $k$ queries and answers of $\mathsf{ADA}_{n,\ell,\mathcal{X}}[\mathsf{M},\mathsf{A}]$ in Step 3b),

2. $(Q_i(\mathcal{U}_T), Y_i)_{i=k+1}^\ell|_E \equiv (Q_i'(\mathcal{U}_{T'}), Y_i')_{i=k+1}^\ell|_{E'}$ (Conditioned on $E$, $\mathsf{A}_2$ successfully reconstruct the master public key $\mathsf{mpk}$ in Step 1b. This yields that conditioned on $E'$ in $\mathsf{ADA}_{n,\ell-k,[m]}[\widetilde{\mathsf{M}}',\widetilde{\mathsf{A}}]$, $\widetilde{\mathsf{M}}'$ perfectly emulates an execution of $\mathsf{ADA}_{n,\ell,\mathcal{X}}[\mathsf{M},\mathsf{A}]$ conditioned on $E$), and

3. $(Q_i'(\mathcal{U}_{T'}), Y_i')_{i=k+1}^\ell \equiv (\widetilde{Q}_i(\mathcal{U}_{[m]}), \widetilde{Y}_i)_{i=1}^{\ell-k}$ ($\widetilde{\mathsf{M}}'$ defines each $q_{i+k}$ by encrypting all the outputs of $\tilde{q}_i$, so in the execution of $\widetilde{\mathsf{M}}'$, for every $i \in [\ell-k]$ it always holds that $q_{i+k}(\mathcal{U}_{\mathcal{T}}) = \tilde{q}_i(\mathcal{U}_{[m]})$, and it also holds that $y_{i+k} = \tilde{y}_i$ by Step 4e).

Hence, we conclude that

$$\Pr\big[\mathsf{ADA}_{n,\ell,\mathcal{X}}[\mathsf{M},\mathsf{A}]=1\big]$$
$$= \Pr[\exists i \in [\ell] \text{ s.t. } |Q_i(\mathcal{U}_T)-Y_i| > 1/10]$$
$$= \Pr[\exists i \in \{k+1,\ldots,\ell\} \text{ s.t. } |Q_i(\mathcal{U}_T)-Y_i| > 1/10 \mid E] \cdot \Pr[E] + 1 \cdot \Pr[\neg E]$$
$$= \Pr\big[\exists i \in \{k+1,\ldots,\ell\} \text{ s.t. } \big|Q_i'(\mathcal{U}_{T'})-Y_i'\big| > 1/10 \mid E'\big] \cdot \Pr\big[E'\big] + \Pr\big[\neg E'\big]$$
$$\geq \Pr\big[\exists i \in \{k+1,\ldots,\ell\} \text{ s.t. } \big|Q_i'(\mathcal{U}_{T'})-Y_i'\big| > 1/10\big]$$
$$= \Pr\Big[\exists i \in [\ell-k] \text{ s.t. } \Big|\widetilde{Q}_i(U_{[m]})-\widetilde{Y}_i\Big| > 1/10\Big]$$
$$= \Pr\Big[\mathsf{ADA}_{n,\ell-k,[m]}[\widetilde{\mathsf{M}}',\widetilde{\mathsf{A}}]=1\Big],$$

as required. The third equality holds by Items 1 and 2 and the penultimate one holds by Item 3.

□

**Algorithm 3.4** (Natural mechanism $\widetilde{\mathsf{M}}$)**.**

**Public parameters:** *Number of samples* $1^n$, *number of queries* $1^\ell$, *and a domain* $\widetilde{\mathcal{X}} = [m]$ *for* $m = 2000n$.

**Oracle Access:** $\widetilde{\mathsf{M}}$ *has access to an IBE scheme* $\mathcal{E} = (\mathsf{Setup}, \mathsf{KeyGen}, \mathsf{Encrypt}, \mathsf{Decrypt})$ *that supports $m$ identities with security parameter $\lambda = \lambda(n)$. Let $k = k(n)$ be the sizes of the keys in this scheme.*

**Setting:** $\widetilde{\mathsf{M}}$ *has access to a mechanism $\mathsf{M}$ and to algorithms $\mathsf{A}_1$ and $\mathsf{A}_2$ (Algorithms 3.1 and 3.2, respectively) and interacts in* $\mathsf{ADA}_{n,\ell,[m]}[\cdot, \widetilde{\mathsf{A}}]$ *(Game 2.7), where* $\widetilde{\mathsf{A}} = (\widetilde{\mathsf{A}}_1, \widetilde{\mathsf{A}}_2)$ *is the pair of algorithms from Theorem 2.8.*

**Operation:**

1. *% Step 1 of Game 2.7: Receive $\mathcal{J} \leftarrow [m]^n$ from $\widetilde{\mathsf{A}}_1$.*

2. *Sample* $(\mathsf{mpk}, \mathsf{msk}) \sim \mathsf{Setup}(1^\lambda)$ *and* $\mathsf{sk}_j \sim \mathsf{KeyGen}(\mathsf{msk}, j)$ *for each* $j \in [m]$.

3. *Start an emulation of $\mathsf{M}$ in the game* $\mathsf{ADA}_{n,\ell+k,\mathcal{X}}[\cdot, \mathsf{A}]$ *for* $\mathcal{X} = [m] \times \{0,1\}^{2k}$, *where:*

   (a) *In Step 1 of the emulation, let $\mathsf{M}$ receive the samples* $\mathcal{S} = \{(j, \mathsf{mpk}, \mathsf{sk}_j)\}_{j \in \mathcal{J}}$ *which plays the role of $n$ i.i.d. samples from* $\mathcal{T} = \{(j, \mathsf{mpk}, \mathsf{sk}_j)\}_{j \in [m]}$ *(i.e., the $n$ samples that $\mathsf{A}_1$ sends to $\mathsf{M}$ in the emulation).*

   (b) *Emulate the first $k$ queries and answers $q_1, y_1, \ldots, q_k, y_k$ when interacting with $\mathsf{A}_2$ in* $\mathsf{ADA}_{n,\ell+k,\mathcal{X}}[\mathsf{M}, \mathsf{A}]$ *(Step 1 of Algorithm 3.2).*

4. *% Step 2 of Game 2.7:*

   *For $i = 1, \ldots, \ell$:*

   (a) *% Step 2a: Receive a query $\tilde{q}_i \colon [m] \to \{-1, 0, 1\}$ from $\widetilde{\mathsf{A}}_2$.*

   (b) *For $j \in \mathcal{J}$ compute* $\mathsf{ct}_{i,j} \sim \mathsf{Encrypt}(\mathsf{mpk}, j, \tilde{q}_i(j))$ *and for $j \in [m] \setminus \mathcal{J}$ compute* $\mathsf{ct}_{i,j} \sim \mathsf{Encrypt}(\mathsf{mpk}, j, 0)$.

   (c) *Continue the emulation of $\mathsf{ADA}_{n,\ell+k,\mathcal{X}}[\mathsf{M}, \mathsf{A}]$ by sending $\{\mathsf{ct}_{i,j}\}_{j=1}^{m}$ to $\mathsf{M}$ as the $(k+i)$'th query $q_{i+k}$ of $\mathsf{A}_2$.*

   (d) *Let $y_{k+i}$ be the answer that $\mathsf{M}$ sends in the emulation (in response to the $(k+i)$'th query).*

   (e) *% Step 2b: Send the answer $\tilde{y}_i = y_{k+i}$ to $\widetilde{\mathsf{A}}_2$.*

# 4 Reduction to Natural Mechanisms Implies Key Agreement

In this section we prove that any PPT *balanced* adversary $\mathsf{A} = (\mathsf{A}_1, \mathsf{A}_2)$ that has the structure of all known lower bounds ([HU14; SU15b; DSWZ23] and ours in Section 3), can be used to construct a *key-agreement* protocol.

All known constructions use an adversary $\mathsf{A}$ that wraps the adversary $\widetilde{\mathsf{A}}$ for the natural mecha-

nisms case (Theorem 2.8) by forcing every mechanism M to behave *naturally* using cryptography. In particular, they all have the following two key properties that are inherited from $\widetilde{\mathsf{A}}$:

1. The analyst asks queries that it knows the true answer to them (i.e., the true answer can be extracted from its view), and

2. If a PPT mechanism attempts to behave accurately (e.g., given a query, it answers the empirical mean), then in the last round it will fail with high probability (which is the analog of Observation 2.9).

The formal statement is given in the following theorem.

**Theorem 4.1** (Restatement of Theorem 1.6)**.** *Assume the existence of a* PPT *adversary* $\mathsf{A} = (\mathsf{A}_1, \mathsf{A}_2)$ *and functions* $\ell = \ell(n) \leq \operatorname{poly}(n)$ *and* $\mathcal{X} = \mathcal{X}(n)$ *with* $\log|\mathcal{X}| \leq \operatorname{poly}(n)$ *such that the following holds: Let* $n \in \mathbb{N}$ *and consider a random execution of* $\mathsf{ADA}_{n,\ell,\mathcal{X}}[\mathsf{M}, \mathsf{A}]$ *where* M *is the mechanism that given a sample* $\mathcal{S}$ *and a query* $q$*, answers the empirical mean* $q(\mathcal{S})$*. Let* $D_n$ *and* $Q_n$ *be the (r.v.'s of the) values of* $\mathcal{D}$ *and* $q = q_\ell$ *(the last query) in the execution (respectively), let* $T_n$ *be the transcript of the execution between the analyst* $\mathsf{A}_2$ *and the mechanism* M *(i.e., the queries and answers), and let* $V_n$ *be the view of* $\mathsf{A}_2$ *at the end of the execution (without loss of generality, its input, random coins and the transcript). Assume that*

1. $\exists$ PPT *algorithm* F *s.t.* $\forall n \in \mathbb{N}: \Pr\left[|\mathsf{F}(V_n) - Q_n(D_n)| \leq n^{-1/10}\right] \geq 1 - \operatorname{neg}(n)$, *and*

2. $\forall$ PPT *algorithm* G *and* $\forall n \in \mathbb{N}: \Pr[|\mathsf{G}(T_n) - Q_n(D_n)| \leq 1/10] \leq 1/4 + \operatorname{neg}(n)$.

*Then using* A *and* F *it is possible to construct a fully-secure key-agreement protocol.*

Note that Assumption 1 in Theorem 4.1 formalizes the first property in which the analyst knows a good estimation of the true answer, and the PPT algorithm F is the assumed knowledge extractor. Assumption 2 in Theorem 4.1 formalizes the second property which states that the mechanism, which answers the empirical mean along the way, will fail in the last query, no matter how it chooses to act (this behavior is captured with the PPT algorithm G), and moreover, it is enough to assume that this requirement only holds with respect to to transcript of the execution, and not with respect to the view of the mechanism.

**Example: Our Adversary from Section 3** In the following we show that our adversary $\mathsf{A} = (\mathsf{A}_1, \mathsf{A}_2)$ (Algorithms 3.1 and 3.2) has the above properties. Using similar arguments it can be shown that any previously known lower bound [HU14; SU15b; DSWZ23] has these properties as well.

Recall that the *sampler* $\mathsf{A}_1$ first samples keys $\mathsf{mpk}, \mathsf{msk}, \{\mathsf{sk}_j\}_{j=1}^m$ ($m = 2000n$), and generates $n$ uniformly random samples from the triplets $\mathcal{T} = \{(j, \mathsf{mpk}, \mathsf{sk}_j)\}_{j=1}^m$. The mechanism M gets the samples, and assume that M simply outputs the empirical mean of each query. Therefore, in the first $k$ queries of the interaction, the analyst $\mathsf{A}_2$ discovers the master key $\mathsf{mpk}$. This allows it to wrap each query $\tilde{q}_i$ of the analyst $\widetilde{\mathsf{A}}_2$ by encrypting all the values using $\mathsf{mpk}$, and sending the encryptions to M. Therefore, it is clear that $\mathsf{A}_2$ knows the true mean for each wrapped query $q_i$ (and in particular, the last one), since it equals to $\frac{1}{m} \sum_{j=1}^m \tilde{q}_i(j)$ (a description of $\tilde{q}_i$ is part of the view of $\mathsf{A}_2$). This fulfills Assumption 1 of Theorem 4.1.

Regarding Assumption 2, note that when M simply answers the empirical means along the way, it is translated to answering the empirical means to the analyst $\widetilde{A}_2$. Therefore, by Observation 2.9, $\widetilde{A}_2$ will provide a last query that fails each last round strategy. By the properties of the IBE scheme, M does not see any values beyond its $n$ samples, which forces it to behave like a *natural* mechanism. In particular, the above also holds when M uses a last round strategy G which is only a function of the transcript (which is only part of the view of M).

## 4.1 Proving Theorem 4.1

Theorem 4.1 is an immediate corollary of the following Lemma 4.3 and Theorem 2.4.

---

**Protocol 4.2** (Approximate agreement protocol $(P_1, P_2)$).

**Input:** *A security parameter $1^n$. Let $\ell = \ell(n)$ and $\mathcal{X} = \mathcal{X}(n)$ be as in Theorem 4.1.*

**Access:** *Each $P_i$ , for $i \in [2]$, has access to algorithm $A_i$ from Theorem 4.1. $P_2$ has also access to algorithm F from Theorem 4.1.*

**Operation:**

- $P_1$ *emulates $A_1$ on input $1^n$, $1^\ell$, and $\mathcal{X}$ for obtaining a distribution $\mathcal{D}$ (specified by a sampling procedure), and then samples $2n$ i.i.d. samples according to it. Let $\mathcal{S}$ be the first $n$ samples, and let $\mathcal{S}'$ be the last $n$ samples.*

- $P_2$ *initializes an emulation of $A_2$ on inputs $1^n$, $1^\ell$, and $\mathcal{X}$.*

- *For $i = 1$ to $\ell$:*

  1. $P_2$ *receive the $i^{\text{th}}$ query $q_i$ from the emulated $A_2$ and send it to $P_1$.*
  2. $P_1$ *sends $y_i = q_i(\mathcal{S})$ to $P_2$.*
  3. $P_2$ *sends $y_i$ as the $i^{\text{th}}$ answer to the emulated $A_2$.*

- $P_1$ *outputs $q_\ell(\mathcal{S}')$.*

- $P_2$ *outputs $F(v)$ where $v$ is the view of $A_2$ in the emulation.*

---

**Lemma 4.3.** *Let $A = (A_1, A_2)$, $\ell = \ell(n)$, $\mathcal{X} = \mathcal{X}(n)$, and F be as in Theorem 4.1. Then Protocol 4.2 (w.r.t. these values) is an $(2 \cdot n^{-1/10}, 1/20)$-approximate agreement protocol according to Definition 2.3.*

*Proof.* Consider a random execution of $(P_1, P_2)$ on input $1^n$. Let $D_n, S_n, S'_n, Q_n, V_n$ be the values of $\mathcal{D}, \mathcal{S}, \mathcal{S}', q_\ell, v$ (respectively), and let $O_n^1 = Q(S'_n)$ and $O_n^2 = F(V)$ be the outputs of $P_1$ and $P_2$ (respectively). Let $E$ be the event $|Q_n(S'_n) - Q(D_n)| \leq n^{-1/10}$. Note that $Q_n$ and $S'_n$ are independent conditioned on $D_n$, and $S'_n$ contains $n$ i.i.d. samples from $D_n$. Hence by Hoeffding's inequality (Fact 2.1) it holds that $\Pr[E] \geq 1 - \text{neg}(n)$.

The agreement guarantee holds by the following computation.

$$\Pr\Big[\big|O_n^1 - O_n^2\big| \leq 2 \cdot n^{-1/10}\Big] = \Pr\Big[\big|Q_n(S_n') - \mathsf{F}(V)\big| \leq 2 \cdot n^{-1/10}\Big]$$

$$\geq \Pr\Big[\big|Q_n(S_n') - \mathsf{F}(V)\big| \leq 2 \cdot n^{-1/10} \mid E\Big] \cdot \Pr[E]$$

$$\geq \Pr\Big[\big|Q_n(D_n) - \mathsf{F}(V)\big| \leq n^{-1/10} \mid E\Big] \cdot \Pr[E]$$

$$\geq \Pr\Big[\big|Q_n(D_n) - \mathsf{F}(V)\big| \leq n^{-1/10}\Big] - \Pr[\neg E]$$

$$\geq 3/4 - \mathrm{neg}(n),$$

where the last inequality holds by Assumption 1.

For the secrecy guarantee, note that the transcript $T_n$ between $\mathsf{P}_1$ and $\mathsf{P}_2$ only consists of the transcript between the analysis and the mechanism in $\mathsf{ADA}_{n,\ell,\mathcal{X}}[\mathsf{M},\mathsf{A}]$ where $\mathsf{M}$ is the mechanism that answers the empirical mean of each query (holds by the answers that $\mathsf{P}_1$ sends in Step 2). Therefore, for every PPT adversary $\mathsf{G}$ we conclude that

$$\Pr\big[\big|\mathsf{G}(T_n) - O_n^1\big| \leq 1/20\big] \leq \Pr\big[\big|\mathsf{G}(T_n) - Q_n(S_n')\big| \leq 1/20 \mid E\big] \cdot \Pr[E] + \Pr[\neg E]$$

$$\leq \Pr[|\mathsf{G}(T_n) - Q_n(D_n)| \leq 1/10 \mid E] \cdot \Pr[E] + \Pr[\neg E]$$

$$\leq \Pr[|\mathsf{G}(T_n) - Q_n(D_n)| \leq 1/10] + \Pr[\neg E]$$

$$\leq 1/4 + \mathrm{neg}(n),$$

as required. The last inequality holds by Assumption 2.

$\square$

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

# A   Proving Theorem 2.4

Theorem 2.4 is an immediate corollary of the following statements.

**Theorem A.1** (Key agreement amplification, a corollary of Theorem 7.5 in [Hol06]). *If there exists an $(1 - n^{-\Omega(1)}, 1 - \Omega(1))$-key agreement protocol, then there exists a fully-secure key-agreement protocol.*

**Lemma A.2** (From approximate agreement to a weak key-agreement). *Let $\alpha, \beta \colon \mathbb{N} \to [0,1]$ be efficiently computable functions such that $\alpha(n)/\beta(n) \leq n^{-\Omega(1)}$ and $\alpha(n) \cdot \beta(n) \geq 2^{-n}$ for large enough $n$. If there exists an $(\alpha, \beta)$-approximate-agreement protocol, then there exists an $(1 - n^{-\Omega(1)}, 1 - \Omega(1))$-key-agreement protocol.*

## A.1   Proving Lemma A.2

In the following, for two binary vectors $x, y \in \{0,1\}^m$, we let $\langle x, y \rangle = \sum_{i=1}^{m} x_i y_i$ (the inner product of $x$ and $y$), and let $x \oplus y = (x_1 \oplus y_1, \ldots, x_m \oplus y_m)$ (i.e., the bit-wise XOR of $x$ and $y$). To prove Lemma A.2, we use the following weak version of Goldreich Levin [GL89].

**Lemma A.3.** *There exists a* PPT *oracle-aided algorithm* Dec *such that the following holds for every* $n \in \mathbb{N}$. *Let* $m \leq n$, $x \in \{0,1\}^m$ *and* A *be an algorithm such that*

$$\Pr_{r \sim \{0,1\}^m}[\mathsf{A}(r) = \langle x, r \rangle \bmod 2] > 3/4 + 0.01.$$

*Then* $\Pr\left[\mathsf{Dec}^\mathsf{A}(1^n, 1^m) = x\right] \geq 1 - \mathrm{neg}(n)$.

*Proof.* We use A to decode each bit of $x$ separately. For every $i$, let $e_i \in \{0,1\}^m$ be the vector that has 1 in the $i^{\text{th}}$ entry and 0 everywhere else. Observe that

$$\Pr_{r \leftarrow \{0,1\}^m}[\{\mathsf{A}(r) = \langle x, r \rangle \bmod 2\} \wedge \{\mathsf{A}(r \oplus e_i) = \langle x, r \oplus e_i \rangle \bmod 2\}]$$
$$\geq 1 - 2 \cdot \Pr_{r \leftarrow \{0,1\}^m}[\{\mathsf{A}(r) \neq \langle x, r \rangle \bmod 2\}]$$
$$\geq 1/2 + 0.01$$

By the linearity of the inner product we deduce that,

$$\Pr_{r \leftarrow \{0,1\}^m}[\mathsf{A}(r) \oplus \mathsf{A}(r \oplus e_i) = x_i] \geq 1/2 + 0.01.$$

Let Dec be the algorithm that for every $i$, computes $\mathsf{A}(r) \oplus \mathsf{A}(r \oplus e_i)$ for $n$ random values of $r \leftarrow \{0,1\}^m$, and let $x'_i$ be the majority of the outputs. Then Dec outputs $x' = (x'_1, \ldots, x'_n)$. By Hoeffding's inequality Fact 2.1, each $x'_i$ is equal to $x_i$ with all but $e^{-\Omega(n)}$ probability. Since $m \leq n$ we conclude by the union bound that the above is true for all $i$'s simultaneously with all but $\mathrm{neg}(n)$ probability, as required.

$\square$

We now ready to prove Lemma A.2 that transforms an approximate-agreement protocol into a weak key-agreement protocol.

---

**Protocol A.4** (Weak key-agreement protocol $(\mathsf{P}_1, \mathsf{P}_2)$).

*Input:* $1^n$.

*Access:* An $(\alpha, \beta)$-approximate-agreement protocol $(\mathsf{P}'_1, \mathsf{P}'_2)$.

*Operation:*

- *Let* $\gamma = \sqrt{\alpha(n)\beta(n)}$, *and let* $\mathcal{B} = \{-1, -1+\gamma, -1+2\gamma, \ldots, -1 + \lfloor 2/\gamma \rfloor \cdot \gamma\}$ *be a division of* $[-1, 1]$ *into buckets, each can be represented using* $m = \lceil \log_2(2/\gamma) \rceil$ *bits.*

- *The parties (jointly) emulate* $(\mathsf{P}'_1, \mathsf{P}'_2)$ *on input* $1^n$, *where each* $\mathsf{P}_i$ *takes the role of* $\mathsf{P}'_i$. *Let* $o'_i$ *be the output of the emulated* $\mathsf{P}'_i$.

- $\mathsf{P}_1$ *chooses* $v \leftarrow [0, \gamma]$ *and* $r \leftarrow \{0,1\}^m$ *and sends them to* $\mathsf{P}_2$.

- *Each* $\mathsf{P}_i$ *computes* $s_i \in \{0,1\}^m$ *as the binary representation of the bucket* $b_i = \mathrm{argmin}_{b \in \mathcal{B}}\{|b - (o'_i + v)|\}$, *and (locally) outputs* $o_i = \langle s_i, r \rangle \bmod 2$.

*Proof of Lemma A.2.* Let $(\mathsf{P}_1', \mathsf{P}_2')$ be an $(\alpha, \beta)$-approximate-agreement protocol where $\alpha(n) \cdot \beta(n) \geq 2^{-n}$ and $\alpha(n)/\beta(n) \leq n^{-c}$ for a constant $c > 0$ and large enough $n$. We prove the lemma by showing that Protocol A.4 $(\mathsf{P}_1, \mathsf{P}_2)$ is an $(1 - n^{-c/2}, 0.9)$-key-agreement protocol.

Fix large enough $n \in \mathbb{N}$ and consider a random execution of $(\mathsf{P}_1, \mathsf{P}_2)(1^n)$. Let $O_1'$, $O_2'$, $V$, $R$, $B_1$, $B_2$, $O_1$, $O_2$ be the (r.v.'s of the) values of $o_1'$, $o_2'$, $v$, $r$, $b_1$, $b_2$, $o_1$, $o_2$ in the execution, let $T$ be the transcript of the execution, and let $T'$ be the transcript of the emulated execution $(\mathsf{P}_1', \mathsf{P}_2')(1^n)$ in Step A.4. By the approximate agreement property of $(\mathsf{P}_1', \mathsf{P}_2')$, it holds that

$$\Pr\left[|O_1' - O_2'| \leq \alpha\right] \geq 1 - \mathrm{neg}(n) \tag{3}$$

Since $V$ is independent of $O_1'$ and $O_2'$, it holds that

$$\Pr\left[B_1 = B_2 \mid |O_1' - O_2'| \leq \alpha\right] \geq 1 - \alpha/\gamma \geq 1 - n^{-c/2} \tag{4}$$

Hence, the agreement of $(\mathsf{P}_1, \mathsf{P}_2)$ holds by the following computation.

$$\begin{aligned}
\Pr[O_1 = O_2] &\geq \Pr[B_1 = B_2] \\
&\geq \Pr\left[B_1 = B_2 \mid |O_1' - O_2'| \leq \alpha\right] \cdot \Pr\left[|O_1' - O_2'| \leq \alpha\right] \\
&\geq (1 - n^{-c/2}) \cdot (1 - \mathrm{neg}(n)) \\
&= 1 - n^{-c/2} - \mathrm{neg}(n)
\end{aligned}$$

In order to prove the secrecy of $(\mathsf{P}_1, \mathsf{P}_2)$, assume towards a contradiction that there exists a PPT $\mathsf{A}$ such that

$$\Pr[\mathsf{A}(T) = O_1] \geq 0.9 \tag{5}$$

By Lemma A.3, there exists a PPT oracle-aided algorithm $\mathsf{Dec}$ such that

$$\Pr\left[\mathsf{Dec}^{\mathsf{A}}(T) = S_1\right] \geq 1 - \mathrm{neg}(n). \tag{6}$$

Let $\mathsf{A}'$ be the PPT algorithm that given a transcript $t'$ of the execution of $(\mathsf{P}_1', \mathsf{P}_2')$, samples $v \leftarrow [0, \gamma]$ and $r \leftarrow \{0,1\}^m$ (as in Step A.4 of Protocol A.4), computes $t = (v, r, t')$ and outputs the bucket $b \in \mathcal{B}$ that is represented by the binary string $\mathsf{Dec}^{\mathsf{A}}(t) \in \{0,1\}^m$. Since the transcript $t$ induced by $\mathsf{A}'(T')$ is distributed the same as $T$, we conclude that

$$\begin{aligned}
\Pr\left[|\mathsf{A}'(T') - O_1'| \leq \beta\right] &\geq \Pr\left[|\mathsf{A}'(T') - O_1'| \leq 2\gamma\right] \\
&\geq \Pr\left[|\mathsf{A}'(T') - O_1'| \leq 2\gamma \mid |O_1' - O_2'| \leq \alpha\right] - \mathrm{neg}(n) \\
&\geq \Pr\left[\mathsf{Dec}^{\mathsf{A}}(T) = S_1 \mid |O_1' - O_2'| \leq \alpha\right] - \mathrm{neg}(n) \\
&\geq \Pr\left[\mathsf{Dec}^{\mathsf{A}}(T) = S_1\right] - \mathrm{neg}(n) \\
&\geq 1 - \mathrm{neg}(n),
\end{aligned}$$

in contradiction to the secrecy property of $(\mathsf{P}_1', \mathsf{P}_2')$. $\qquad \square$