# OpenReview forum: "Adaptive Data Analysis in a Balanced Adversarial Model"
_NeurIPS.cc/2023/Conference — NeurIPS 2023 spotlight_

### Official Review · Reviewer_Ejfp · 2023-07-04

**Soundness:** 2 fair
**Presentation:** 2 fair
**Contribution:** 3 good
**Rating:** 6
**Confidence:** 2

**Summary:**

The paper considers balanced adversaries that do not have access to the underlying distribution and instead interact with an oracle adversary that chooses the distribution and samples points from it. The authors show that with this more realistic adversarial setting, they can improve the previous known lower bounds for adaptive data analysis.

**Strengths:**

The authors consider a more realistic balanced adversary that does not have access to the underlying distribution. This adversarial setting is novel and allows for better bounds on the adaptive data analysis.

**Weaknesses:**

While considering this balanced adversary is more practical, I would suggest the authors to include a summary table comparing their setting, assumptions and their bounds with respect to the prior works. This would better highlight the differences and focus more on the key contribution of the paper.



**Questions:**

I am confused why the authors have the value 2000n in Theorem 2.4. Is there something particular about this value 2000, and if so how do they arrive at this? Can this be a parameter k where k does not take large values?

**Limitations:**

I don't feel there is any direct or evident negative societal impact of this work.

---

> ### Author Rebuttal · Authors · 2023-08-06
>
> Thank you for reviewing our paper. In the following we would like to respond to the specific points you make:
>
> Regarding the weakness:
> Thank you for the suggestion. We will add a comparison table.
>
> Regarding your question:
> We are following Steinke and Ullman [2015a] who used 2000n as a large enough domain size that allows their attack against natural mechanisms to work. For the theoretical result, the constant 2000 is not important; any large enough constant would work (see Sections 3.3-3.4 in their arxiv version - CoRR, abs/1410.1228). The same constant is also used in other works that rely on their lower bound (e.g., see Algorithm 4 in Dinur et al. [2023] - CoRR, abs/2302.05707).

---

> > ### Comment · Reviewer_Ejfp · 2023-08-16
> >
> > Thank you for the clarifications.

---

### Official Review · Reviewer_CEqx · 2023-07-04

**Soundness:** 4 excellent
**Presentation:** 3 good
**Contribution:** 4 excellent
**Rating:** 9
**Confidence:** 5

**Summary:**

Adaptive data analysis considers the setting where a mechanism and an analyst play a game against each other, where for $k$ sequential rounds, the analyst issues queries to a mechanism which holds a dataset of size $n$ sampled from a distribution, the mechanism returns a response, and the analyst chooses the next query accordingly. The mechanism wins the game if all responses are close to the expected value of the queries on the dataset.

Using tools developed in the context of differential privacy, it was proven that a mechanism can respond to up to $k = O(n^{2})$ queries, while winning the game with high probability. At the same time, it was proven that if the number of queries exceeds $O(n^{2})$, there exists an analyst that can win the game. This lower bound assumes the analyst can choose the distribution, and is based on the cryptographic assumption of the existence of fingerprinting codes. This imposes an asymmetry between the power and knowledge of the analyst and mechanism which is unnatural, since an analyst that knows the distribution has no reason to query the mechanism.

The current work solves this problem, by proving a similar lower bound for a balanced setting, where the distribution is chosen by a third player. This proof is based on the stronger cryptographic assumption of the existence of public-key cryptography. The authors also prove that the lower bound does not hold, without this assumption, proving a separation between the balanced and unbalanced settings.

**Strengths:**

The deep asymmetry that was used in the proof the previous lower bounds was undesired, and left an open question whether it is essential, since it is not justified in most natural settings. Solving this open question is an important step in understanding the dynamics of adaptive data analysis. They are based on a novel addition to the existing lower bound framework, and are presented in a clear and formal way.

The distinction between the two cryptographic assumptions required in both cases, might hold implications for cryptography as well.

**Weaknesses:**

I have only one minor comment to add. It seems like the authors use another implicit assumption throughout the paper, that the mechanism knows the analyst it is playing against. If this was not the case, the separation of the analyst into two analysts (one choosing the distribution and the other issuing the queries) will be meaningless, as we could always consider the case where the first analyst always chooses the same distribution, and so the second one can use that information while the mechanism cannot.

I do not bring this up as a limitation of the work, but as a recommendation to clarify the setting, so to avoid confusion by the reader.

=======

**Edit after rebuttal discussion:**

I have no further concerns.

**Questions:**

In section 2.4 the authors mention the sample size, number of queries and dimensions are provided in unary representation. I would appreciate if the authors can clarify how this affects the computational complexity, and how they justify this decision.

---

> ### Author Rebuttal · Authors · 2023-08-06
>
> We thank you for your positive review. In the following we would like to respond to specific points you make.
>
> Regarding the weakness:
>
> In our model, the sampler $A_1$ is publicly known (similar to the Bayesian ADA model of Elder [2016] as we note in Section 1.2). This way, $A_1$ captures knowledge about the underlying distribution which is known to both the mechanism $M$ and the analyst $A_2$. In principle, it suffices that the lowerbound would hold even if the code of the analyst $A_2$ is not publicly known. However, our lower bound in Thm 1.5 holds even if $A_2$ is publicly known (this makes it a stronger bound). We plan to include text to clarify this point.
>
> Regarding the question:
>
> The unary representation is commonly used in the cryptography community to allow algorithms run in polynomial time in a parameter. We basically want to say that an algorithm $A$, given ($n,\ell,d$), runs in poly($n,\ell,d$) time (i.e., that $A$ is a PPT algorithm). But the binary description of $n,\ell,d$ is of logarithmic size in $n, \ell, d$. The unary representation is a convenient way to allow $A$ run in time polynomial in the values $n,\ell,d$.

---

> > ### Comment · Reviewer_CEqx · 2023-08-15
> >
> > I thank the reviewers for the clarification, and for the opportunity to learn about this exciting result.

---

### Official Review · Reviewer_Mqg2 · 2023-07-05

**Soundness:** 3 good
**Presentation:** 2 fair
**Contribution:** 2 fair
**Rating:** 5
**Confidence:** 2

**Summary:**

The paper proposes to consider balanced adversaries for ADA. Under the public-key assumptions, the paper proves that

**Strengths:**

1. The paper assumes a more practical threat model for ADA.

2. The theoretical results in this paper seem to be well-supported. I am not an expert in this research direction, so I am not sure about my judgement.



**Weaknesses:**

1. There is no discussion about the real-world machine learning applications of the results.

2. There is no conclusion section to summarize the paper. The paper ends after Section 4 and makes me feel that the paper is not complete.

3. The theoretical results are based on the assumption that public-key cryptography exists.

4. I am not an expert in this research direction, and I feel that this paper is not easy to follow because there are many terms. I suggest that the paper could add a table to explain the terms used in the paper to improve the readability.


**Questions:**

1. Is there any real-world machine learning application of the results?

---

> ### Author Rebuttal · Authors · 2023-08-06
>
> Thank you for reviewing our paper. We will implement your suggestions to make our paper more broadly accessible.
>
> Statistical queries are basic building blocks in ML and statistics. Our result is about the possibility of accurately answering adaptively chosen statistical queries and hence has consequences to ML and statistics.
>
> The problem we are addressing is not new and our lower bound improves over previously known bounds. Overall, our and previous bounds show that it is impossible to construct efficient mechanisms that answer more than $n^2$ queries adaptively. However, prior work left open the possibility that such mechanisms exist when the underlying distribution is not chosen by the analyst. Our lower bound closes this gap.
>
> Lower bounds such as ours show that no general solution exists for a problem. They often use unnatural inputs or distributions and rely on cryptographic assumptions (our lower bound relies on a well accepted assumption). They are important as guidance for how to proceed with a problem, e.g., search for mechanisms that would succeed if the underlying distribution is from a "nice" family of distributions.

---

> > ### Comment · Reviewer_Mqg2 · 2023-08-19
> > **Thank you for the rebuttal**
> >
> > Thank you for the clarifications.

---

### Official Review · Reviewer_vTGp · 2023-07-06

**Soundness:** 4 excellent
**Presentation:** 4 excellent
**Contribution:** 3 good
**Rating:** 6
**Confidence:** 4

**Summary:**

The paper studies limitations of "adaptive data analysis" with a focus on "computational assumptions" that are used for proving such limitations (i.e., lower bounds).

A mechanism $M$ answers $m=m(n)$ queries (almost) correctly, if it succeeds in the following game for every distribution $D$ and PPT adversary $A$. The mechanism is given n iid samples from D. Then the adversary $A$ asks $m$ adaptive statistical queries $q_1,\dots,q_m$ from the mechanism $M$. The mechanism succeeds if all of its answers are within $\pm 1/10$ of the true answers with respect to distribution $D$.

Previous work has shown that at most $m=\Theta(n^2)$ queries can be answered adaptively by computationally bounded mechanisms, assuming one-way functions exist. Their proof uses private-key encryption schemes (whose existence is equivalent to one-way functions) and a reduction from a class of mechanisms, called "natural". The previous lower bound also implicitly assumes a rather powerful adversary who both chooses the distribution D as well as the queries $q_1,...q_m$.

This paper starts by making the (interesting) observation that if an adversary already knows the distribution $D$, then it won't have a meaningful interest in making the queries. So, in some sense, the previous lower bounds are not as meaningful as one wishes in real world scenarios; this raises the hope for getting positive results that go around the lower bounds by leveraging this observation.

In summary, the main result of the paper is bootstrap the previous lower bound to a model to a weaker adversarial model.
In more detail, the contributions of this paper are as follow:

1. A new formal model of security called the "balanced model" is defined, in which the adversary is split into two entities that do not communicate. The first adversary $A_1$ picks the distribution $D$ and the iid samples. The second adversary $A_2$ picks the adaptive queries.
2. It is shows that assuming the stronger primitive of secure public-key encryption (as opposed to just OWFs) one can still extend the previous lower to the balanced model.
3. The assumption of public-key encryption is somehow justified, by showing any lower bound with minimal properties (that hold for current lower bound proofs) would lead to key agreement protocols (which is a close primitive/assumption to public key encryption).

**Strengths:**

The main strengths of the paper:

* excellent writing quality. my only criticism is that the definition used in *previous* work is not presented formally, and one cannot determine the exact details from the informal text. this is needed for making a fully detailed comparison between the new definition and the old one, because this is a major part of this paper.

* revisiting a previous lower bound for an important problem from a natural practical perspective.

**Weaknesses:**

My criticisms of the paper are the following:

1. About the model (Item 1 above): I am not sure if the new split (balanced) model is adding much value actually. suppose we fix a mechanism $M$ and look for a balanced 2-part polynomial-size adversary $(A_1,A_2)$ against $M$. In this setting, it seems meaningless to say that only $A_1$ knows the distribution D, because one can fix the distribution $D$ that $A_1$ picks to its "best possible" distribution (against $M$). This way, $A_2$ also would be aware of this distribution implicitly, as it can depend on it. In other words, one can see this argument as giving shared coin to $A_1,A_2$, and then fixing it to its best value (which leads to no communication between $A_1$ and $A_2$).

I shall add that, from a game theoretic perspective, it would seems a preferred and stronger lower bound if we could use *one* fixed adversary strategy that bounds the power of every computationally limited mechanism. However, for the sake of proving lower bounds, it is fine if we'd pick the adversary $A_M$ based on the mechanism $M$.

2. About the technical depth of the lower bound (Item 2 above). This proof seems like a rather straightforward adaptation of the previous proof. but using IBE (or a bunch of public key encryption schemes) instead of Secret Key Encryption schemes. This is a mild criticism, as I am in general in favor of simpler proofs. But this proof seems like a simple proof for a new question rather than a simple proof for a question that existed before, so I am a bit torn here.

3. About necessity of Key Agreement (Item 3 above): I am not a fan of investing in weakening assumptions that are used for *lower bounds*. as I believe there is a fundamental difference between weakening assumptions behind positive results vs those of negative results. When we weaken the assumption behind, say, public key encryption from an assumption $P_1$ to a weaker assumption $P_2$, it means that the schemes that would be deployed *right now* would not break at an unknown time (with potential catastrophic consequences) even if $P_1$ turns to be false. However, when we weaken the (still widely believed) computational assumption behind a a *lower bound*, is is a completely different story, because even using an assumption as strong as IO to prove an impossibility has the same practical consequence as using OWFs. The two scenarios would diverge (and a weaker assumption would start to look interesting) after breaking every IO scheme, which is most likely never gonna happen, and the weakening assumption changes nothing right now. So, in summary, when it comes to lower bounds I am totally fine with using something as strong as any assumption that seems strong enough for the foreseeable future.

update: the issues above were discussed and I am happy with author(s)' response.

**Questions:**

My main question is about the meaningfulness of the balanced attack model which is described above.

**Limitations:**

The limitations are also elaborated in the "weakness" section of the review above.

---

> ### Author Rebuttal · Authors · 2023-08-06
>
> We thank you for the comments. In the following we would like to respond to specific points you make.
>
> W1: In our model, $A_1$ is publicly known. It defines a publicly known prior, similarly to the Bayesian ADA model of Elder [2016] (mentioned in Section 1.2). This way, $A_1$ captures the common knowledge that both the mechanism $M$ and the analyst $A_2$ have about the underlying distribution. We agree that showing that for every $M$ there exists a balanced adversary $A=(A_1,A_2)$ that fails it does not add much value. However, showing that there exists a publicly known balanced adversary $A=(A_1,A_2)$ that fails all mechanisms (as we do) is different and a significant improvement over the previous lower bounds. (In fact, our lower bound in Theorem 1.5 is stronger than what is required for a lower bound as it also uses a publicly known analyst $A_2$ - even though the sampler $A_1$ and the analyst $A_2$ are publicly known and cannot communicate with each other, they fail any computationally bounded mechanism).
> We will add text to clarify the advantage of having $A_1$ publicly known.
>
> W2: The criticism about the lower bounds of Hardt and Ullman [2014] and Steinke and Ullman [2015a] is not new and prior work has attempted at addressing them with only partial success. For example, Elder [2016] (mentioned in section 1.2) considered a similar balanced setting. He proved a bound of $n^4$ for a limited family of mechanisms. Our Thm 1.5 gives a (stronger) bound of $n^2$ that holds for all computationally bounded mechanisms.
>
> W3: We believe that determining the minimal required hardness assumption for a lower bound (in general) is informative. Not necessarily because of immediate practical consequences, but because it can reveal the structure of the problem and help steer the research going forward.
>
> For example, as mentioned in lines 113-119, our result implies that the existence of a certain kind of balanced adversaries that fail any *unbounded* mechanism yields an *information theoretic* key agreement protocol (i.e., where the secrecy holds even if the adversary that sees the transcript is computationally unbounded).  As information theoretic key agreement protocols do not exist, we conclude that such balanced adversaries do not exist either. This is in contrast with the imbalanced model for which we know that adversaries (of the same kind) that fail any unbounded mechanism do exist (Steinke and Ullman [2015a]). This gives the first evidence that there might be a separation between the computational and information theoretic setting under the balanced adversarial model (in contrast with the imbalanced model).
>
> Regarding the definition used in previous works:
> In line 59 we explain that in previous works, the adversary (i.e., the analyst) is the one that chooses the distribution at the outset of Game 1.1. This is equivalent to playing against adversaries $A=(A_1,A_2)$ in the following game: (1) $A_1$ chooses a distribution $\cal{D}$ and sends a state $st$ to $A_2$. (2) $M$ receives $n$ i.i.d. samples from $\cal{D}$, and (3) $M$ and $A_2$ play Game 1.1. We will better explain it in our paper.
> The difference from our model is that $A_1$ is allowed to send a state $st$ to $A_2$ after choosing the distribution. An additional (minor) difference is that we chose in our model to let $A_1$ also provide the i.i.d. samples to $M$. This is only useful for Thm 1.6 as we need there that choosing $\cal{D}$ and sampling from $\cal{D}$ are both computationally efficient (which are simply captured by saying that $A_1$ is computationally bounded). We mentioned this implicit assumption in lines 218-222.

---

> > ### Comment · Reviewer_vTGp · 2023-08-18
> > **Ack**
> >
> > Thanks for the responses. I found them very useful, particularly the one about the use of improving assumptions for lower bounds.
> >
> > It would be very helpful if the authors please do include these discussions in the paper (basically about all the raised points).
> >
> > I'd increase my score.

---

> ### Comment · Reviewer_CEqx · 2023-08-15
>
> If I may add to the authors response. The issue raised by reviewer vTGp resembles the one I raised in my review, and I had to spend some time convincing myself what is the right way to think of it, so it might be the case that a fellow reviewer might find my perspective helpful.  To the best of my understanding does not point to a limitation in the current work, but to a subtle lack of formalism in previous works on this subject. Here is how I have been thinking of it.
>
> Even in the setting where the analyst is all powerful, and has access to and control over everything but the sampled dataset, if the analyst always chooses the same distribution and the mechanism knows which analyst does it compete against, it can provide perfectly accurate private responses, since it effectively has access to the underlying distribution. This limitation can be avoided by adding an assumption about the mechanism's knowledge about the attacker, but this is a dangerous assumption which results in brittle guarantees. Instead, the implicit chosen assumption is that the mechanism knows the analyst, but the analyst uses randomness which prevents the mechanism from leveraging this knowledge.

---

> > ### Comment · Reviewer_vTGp · 2023-08-18
> > **thanks**
> >
> > yes, we raised the same point, and thanks for the clarification.

---

### Decision · Program_Chairs · 2023-09-21

**Decision:**

Accept (spotlight)

**Comment:**

This paper gives a lower bound for answering adaptive statistical queries about a dataset. Prior work has established the lower bound using an adversary that knows the distribution from which the dataset is sampled, making it rather unrealistic. This work introduces a weaker \emph{balanced} adversary which consists of two separated algorithms (a sampler that chooses the distribution and provides the samples to the mechanism, and the analyst who chooses the adaptive queries, but does not have a prior knowledge of the underlying distribution). It extends known lower bounds to this weaker adversary. This result significantly improves our understanding of the hardness of adaptive data analysis.